# SRFT: A Single-Stage Method with Supervised and Reinforcement Fine-Tuning for Reasoning

**Yuqian Fu**[♠♣]*, **Tinghong Chen**[♠♣], **Jiajun Chai**[◇], **Xihuai Wang**[♡], **Songjun Tu**[♠♣],
**Guojun Yin**[◇], **Wei Lin**[◇], **Qichao Zhang**[♠♣], **Yuanheng Zhu**[♠♣]†, **Dongbin Zhao**[♠♣]†
[♠] State Key Laboratory of Multimodal Artificial Intelligence Systems, CASIA
[♣] School of Artificial Intelligence, UCAS    [◇] Meituan    [♡] SJTU
{fuyuqian2022,yuanheng.zhu}@ia.ac.cn

## Abstract

Large language models (LLMs) have achieved remarkable progress in reasoning tasks, yet optimally integrating Supervised Fine-Tuning (SFT) and Reinforcement Learning (RL) remains a fundamental challenge. Through a comprehensive analysis of token distributions, learning dynamics, and integration mechanisms from an entropy-based perspective, we reveal key differences between these paradigms: SFT induces coarse-grained global shifts to policy distributions, while RL performs fine-grained selective optimizations. Our analysis further establishes entropy as a critical indicator of training efficacy. Building on these observations, we introduce **S**upervised **R**einforcement **F**ine-**T**uning (**SRFT**), a single-stage framework that unifies both fine-tuning paradigms through entropy-aware weighting mechanisms. SRFT simultaneously applies SFT and RL to directly optimize LLMs using demonstrations and self-exploration rollouts rather than through two-stage sequential methods. Extensive experiments show that SRFT outperforms zero-RL baselines by **9.0%** on five mathematical reasoning benchmarks and by **10.9%** on three out-of-distribution benchmarks. Moreover, by leveraging demonstration data, SRFT maintains a more stable policy entropy, facilitating sustained policy improvement.

## 1 Introduction

Recent advances in Large Language Models (LLMs) for reasoning (OpenAI, 2025; Guo et al., 2025; Anthropic, 2025) have demonstrated remarkable capabilities in complex problem-solving tasks. Despite these remarkable achievements, fine-tuning strategies for enhancing reasoning capabilities remain an active area of research, presenting both opportunities and challenges.

Initial approaches often treat Supervised Fine-Tuning (SFT) and Reinforcement Learning (RL) as distinct, sequential phases. For instance, SFT might be applied for instruction-following, followed by RL for policy optimization. However, this separation presents several challenges: SFT can lead to models that memorize patterns without developing true reasoning abilities, potentially **overfitting** to the training dataset (Chu et al., 2025; Chen et al., 2025a). Conversely, RL methods, while promising for reward optimization, can be **sample-inefficient**, struggle with effective exploration in vast solution spaces (Schmied et al., 2025), or suffer from issues such as mode collapse, where models repeatedly generate similar, suboptimal outputs (Cai et al., 2025).

Instead of simple sequential approaches, recent work (Yan et al., 2025; Wu et al., 2025; Liu et al., 2025a;b; Chen et al., 2025b; Zhang et al., 2025a) has shown a movement towards integrated frameworks that unify SFT and RL paradigms, or dynamically switch between the two fine-tuning methods during the training process. As shown in Figure 1(a), SFT guides the LLM policy toward demonstration distributions, while RL enables the policy to explore improved solutions in the neighborhood of the base policy distribution. Our illustration demonstrates a key limitation: when the base policy is positioned near a suboptimal policy, the RL online rollouts alone cannot effectively navigate to

*Work was done during an internship at Meituan. †Corresponding authors. ‡ This work was supported in part by the Strategic Priority Research Program of Chinese Academy of Sciences under Grant XDA0480302, in part by National Natural Science Foundation of China under Grants 62136008 and 62293541, and in part by Beijing Nova Program under Grant 20240484514 and Beijing Natural Science Foundation QY25186.

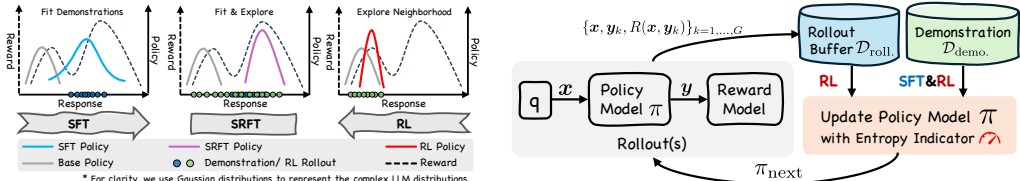

(a) Toy illustration of SFT, RL, and SRFT for LLM reasoning training on a single prompt.

(b) Framework of SRFT. Our method leverages demonstrations to improve reasoning capabilities.

Figure 1: Overview of SRFT's motivation and framework.

the optimal policy. Beyond applying SFT and RL individually, **unified integration of SFT and RL within a single-stage method** (e.g., our proposed method SRFT) enables the LLM policy to **directly optimize toward better solutions across an expanded space**. However, a critical challenge remains in determining the balance between SFT's knowledge distillation and RL's policy optimization: on the one hand, insufficient integration risks error propagation and limits RL improvements; on the other hand, excessive reliance on demonstrations leads to overfitting that constrains exploration beyond the base policy distribution. This trade-off creates uncertainty for practitioners when deciding between SFT, which leverages demonstration data, and RL, which supports online policy exploration.

To address these issues, in this work, we study how to build single-stage LLM fine-tuning algorithms that are not only effective for LLM reasoning from SFT datasets but also well-suited to continuous improvement with RL rollouts. We conduct a comprehensive analysis of the roles that SFT and RL play in fine-tuning LLM reasoning. Through our analysis in Sec. 3, we obtain the following key findings that guide our subsequent algorithm design:

> **Key Findings**
>
> ○ **Policy distribution effects** (Secs. 3.1.1 and 3.1.2): SFT induces coarse-grained global changes to the policy distribution, while RL performs fine-grained selective modifications.
>
> ○ **Single-stage optimization** (Secs. 3.1.2 and 3.2.2): Single-stage integration of SFT and RL enables direct optimization for reasoning capabilities and achieves superior training efficiency compared to sequential SFT→RL approaches.
>
> ○ **Entropy as an indicator** (Sec. 3.2.1): Entropy dynamics reveal the underlying mechanisms of training processes, enabling balanced weighting between the two paradigms.

Based on these insights, we propose **S**upervised **R**einforcement **F**ine-**T**uning (**SRFT**), a single-stage method for LLM reasoning. As shown in Figure 1(b), we integrate SFT into RL and use entropy as a dynamic indicator to **control the balance** between these paradigms. Specifically, for samples generated by LLM policy rollouts, we employ different RL training losses based on whether sample rewards are positive or negative. For samples from demonstration datasets, we simultaneously apply both SFT and RL objectives. This unified approach enables **sample-efficient** learning from demonstrations at multiple granularities while bridging the complementary strengths of SFT and RL. We evaluate our method on five competition-level mathematical reasoning benchmarks and three out-of-distribution benchmarks. SRFT achieves an accuracy of 59.1% based on Qwen-2.5-Math-7B (Yang et al., 2024). Moreover, SRFT demonstrates superior generalization capability, achieving an average improvement of over 4.7% compared to other demonstration-based methods.

Overall, **our key contributions** are:

- We conduct **a comprehensive analysis of SFT and RL in LLM reasoning**, examining their differential effects on LLM policy distributions and learning dynamics. We further analyze the integration of SFT and RL through an entropy-based perspective.

- We propose SRFT, **a single-stage fine-tuning approach** that combines supervised fine-tuning and reinforcement learning with entropy-aware weighting mechanisms, enabling effective utilization of demonstrations while maintaining stable exploration dynamics.

- We demonstrate SRFT's **superior performance across eight challenging benchmarks**, achieving substantial improvements of 9.0% and 10.9% over zero-RL baselines on mathematical reasoning and out-of-distribution tasks, respectively.

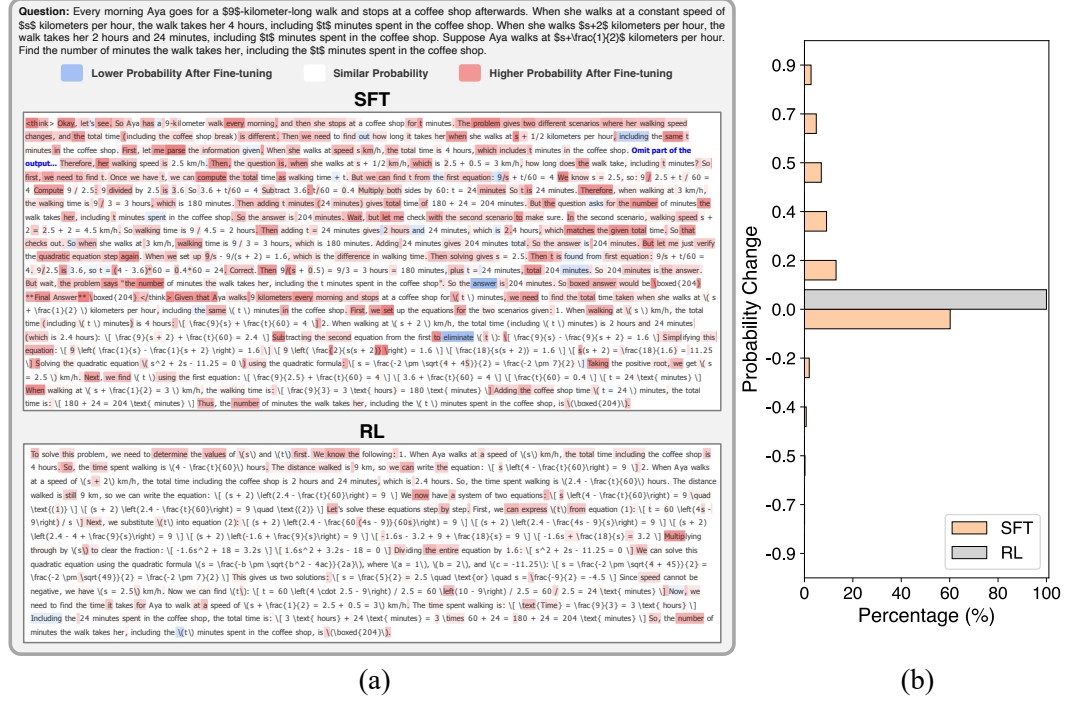

(a)                                    (b)

Figure 2: Visualization of LLM distribution changes during fine-tuning. (a) Heatmap visualization comparing responses generated by fine-tuned and base models, where darker background colors indicate larger probability changes. (b) Distribution of token probability shifts across five benchmarks.

## 2 PRELIMINARIES

**Supervised Fine-Tuning** (SFT) is a standard approach for adapting pre-trained language models to specific downstream tasks. Given a dataset $\mathcal{D} = \{(\boldsymbol{x}_i, \boldsymbol{y}_i)\}_{i=1}^N$, where $\boldsymbol{x}_i$ is an input prompt and $\boldsymbol{y}_i$ is the corresponding target response generated by the behavior policy $\pi_\beta$, the objective is to train the policy $\pi_\theta$ to maximize the conditional probability of generating the target response $\boldsymbol{y}_i$ given $\boldsymbol{x}_i$. This is typically achieved by minimizing the negative log-likelihood over the dataset:

$$\mathcal{L}_{\text{SFT}}(\theta) = \mathbb{E}_{(\boldsymbol{x},\boldsymbol{y})\in\mathcal{D}}[-\log \pi_\theta(\boldsymbol{y}|\boldsymbol{x})]. \tag{1}$$

**Reinforcement Learning** (RL) is typically applied after SFT to further align LLMs with complex behaviors that are challenging to specify through static datasets. To optimize the LLM policy, Group Relative Policy Optimization (GRPO) (Shao et al., 2024) offers a memory-efficient variant of Proximal Policy Optimization (PPO) (Schulman et al., 2017). A key characteristic is that GRPO typically operates without a learned value function. Instead, for a given prompt $\boldsymbol{x}$, it often generates a group of $G$ responses $\{\boldsymbol{y}_1, \ldots, \boldsymbol{y}_G\}$ using the current policy. The rewards $\{R(\boldsymbol{x}, \boldsymbol{y}_1), \ldots, R(\boldsymbol{x}, \boldsymbol{y}_G)\}$ for these responses are then used to compute a relative advantage for each response:

$$\hat{A}_k = \frac{R(\boldsymbol{x}, \boldsymbol{y}_k) - \text{mean}\{R(\boldsymbol{x}, \boldsymbol{y}_k)|k = 1, 2, \ldots, G\}}{\text{std}\{R(\boldsymbol{x}, \boldsymbol{y}_k)|k = 1, 2, \ldots, G\}}. \tag{2}$$

Then, GRPO maximizes a clipped surrogate objective function to ensure stable updates. The objective function for GRPO can then be expressed as:

$$\mathcal{J}_{\text{GRPO}}(\theta) = \frac{1}{G}\sum_{k=1}^{G}\frac{1}{|\boldsymbol{y}_k|}\sum_{t=1}^{|\boldsymbol{y}_k|}\min\left\{r_{k,t}(\theta)\cdot\hat{A}_k, \text{clip}\left(r_{k,t}(\theta), 1-\epsilon, 1+\epsilon\right)\cdot\hat{A}_k\right\}, \tag{3}$$

where $\epsilon$ denotes the clipping range, $\pi_{\theta_{\text{old}}}$ represents the policy before the update, and $r_{k,t}(\theta) = \frac{\pi_\theta(y_{k,t}|(\boldsymbol{x},\boldsymbol{y}_{<t}))}{\pi_{\theta_{\text{old}}}(y_{k,t}|(\boldsymbol{x},\boldsymbol{y}_{<t}))}$ is the importance sampling ratio for each token $y_{k,t}$ in response $\boldsymbol{y}_k$ of length $|\boldsymbol{y}_k|$.

# 3 ANALYSIS OF SFT AND RL IN LLM REASONING

In this section, we provide a comprehensive analysis of the roles of Supervised Fine-Tuning (SFT) and Reinforcement Learning (RL) in LLM reasoning. We first examine their differential effects on token distributions (Sec. 3.1.1) and learning dynamics (Sec. 3.1.2), then investigate the integration mechanisms through an entropy perspective (Sec. 3.2).

## 3.1 SFT AND RL EFFECTS ON LLMS: SLEDGEHAMMER 🔨 VS. SCALPEL 🔪

### 3.1.1 EFFECTS ON TOKEN DISTRIBUTIONS

To understand the differential impact of SFT and RL on reasoning, we visualize token-level probability changes for identical prompts, measured before and after fine-tuning the same base model. As illustrated in Figure 2(a), the results reveal a fundamental difference: SFT substantially reshapes the probability distribution across the entire response sequence, whereas RL primarily adjusts the probabilities of a small subset of tokens, leaving most numerical content and formal proof statements largely unchanged. We further quantify these distribution shifts across five benchmarks, comprising approximately 600,000 token probability samples, as shown in Figure 2(b). The results demonstrate that **SFT induces more pronounced changes in policy distributions compared to RL**. Specifically, token probability shifts from RL are concentrated near zero, whereas SFT leads to changes of a significantly larger magnitude. The details of visualization and further results for other models are provided in Appendices B.1 and I.

From a theoretical perspective, this behavior can be understood through the gradient of the SFT objective function:

$$\nabla_\theta \mathcal{L}_{\text{SFT}} = \mathbb{E}_{(\boldsymbol{x},\boldsymbol{y})\sim\mathcal{D}} \left[ \sum_{t=1}^{|\boldsymbol{y}|} \sum_{v\in\mathcal{V}} \left( \pi_\theta(v|\boldsymbol{x},\boldsymbol{y}_{<t}) - \mathbf{1}_{v=y_t} \right) \nabla_\theta \log \pi_\theta(v|\boldsymbol{x},\boldsymbol{y}_{<t}) \right], \qquad (4)$$

where $\mathcal{V}$ denotes the LLM vocabulary, and $\mathbf{1}_{v=y_t}$ is an indicator function. This formulation reveals that SFT systematically sharpens the model distribution by increasing probabilities for target tokens while decreasing probabilities for all other tokens in the vocabulary, leading to more deterministic outputs. The detailed experiment setting and derivation of Eq. (4) are provided in Appendix B.

### 3.1.2 VISUALIZATION OF LEARNING DYNAMICS

Beyond token-level probability analysis, we investigate the training paradigms through the lens of learning dynamics. Since directly characterizing the LLM feature space is computationally intractable, we introduce a visualization approach that characterizes the evolution of learning dynamics by using the Kullback-Leibler (KL) divergence in policy distribution between a group of existing reference models $\pi_R$ and our fine-tuned model $\pi_\theta$. Specifically, we introduce three reference models: Qwen2.5-Math-7B, DeepSeek-R1, and QwQ-32B. Since it is still challenging to give an exact KL divergence computation over large-scale distributions, we use cross-entropy for its estimation:

$$d_R(\pi_\theta) = H(\pi_R, \pi_\theta) = \mathcal{D}_{\text{KL}}(\pi_R \| \pi_\theta) + H(\pi_R), \quad (5)$$

where $d_R(\pi_\theta)$ denotes the distance between the fine-tuned model $\pi_\theta$ and the reference model $\pi_R$, $H(\pi_R, \pi_\theta)$ is the cross-entropy, and $H(\pi_R)$ is the entropy of the reference model, which is constant for a given reference model. Therefore, $(d_{R_1}, d_{R_2}, d_{R_3})^\mathsf{T}$ serves as a coordinate representing the policy $\pi_\theta$ with the 3-dimensional reference space. As illustrated in Figure 3, all fine-tuning methods improve performance while diverging from the distribution of the base model (Qwen2.5-Math-7B). Notably, SFT induces larger distributional shifts from the initial policy than RL, achieving a +5.0 points performance improvement. This result

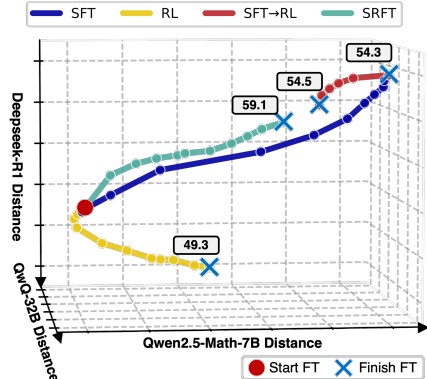

Figure 3: Learning dynamics visualization of different fine-tuning paradigms in probability space. The ⬚number⬚ denotes the final performance of each method.

Table 1: Performance comparison of SFT and RL integration strategies across multiple benchmarks. **Bold** and underlined indicate the best and second-best performance, respectively.

| Model | AIME24 | AMC | MATH500 | Minerva | Olympiad | Avg. |
|---|---|---|---|---|---|---|
| Qwen2.5-Math-7B | 11.4 | 32.6 | 48.8 | 8.7 | 15.8 | 23.5 |
| SFT | 21.2 | 53.2 | 83.0 | 37.1 | 42.2 | 47.3 |
| RL | 21.2 | **59.3** | 83.6 | 36.4 | 46.6 | 49.4 |
| RL→SFT | 10.5 | 40.4 | 73.6 | 32.0 | 30.7 | 37.4 |
| RL→SFT$_{KL}$ | 13.1 | 45.2 | 70.2 | 26.5 | 36.3 | 38.3 |
| SFT→RL | **24.5** | **59.3** | **86.4** | **39.3** | **53.1** | **52.5** |

further validates our findings in Sec. 3.1.1, confirming that SFT drives broader changes, whereas RL fine-tuning tends to remain close to the initial policy. Furthermore, during the SFT→RL pipeline, **the optimization direction in the second-stage RL opposes that of SFT**. This result suggests that SFT may overfit to demonstrations, and the subsequent RL attempts to correct this deviation, thereby **increasing the learning tax** of the two-stage paradigm. By contrast, our single-stage method, SRFT, follows a **more straightforward training trajectory** in probability space, enabling more efficient optimization than sequential fine-tuning approaches. More technical details for this visualization and analysis are provided in Appendix C.

## 3.2 Integration of SFT and RL: From Two-stage 🔗 to Single-stage 🔗

### 3.2.1 Sequential Integration Analysis

In this section, we examine the integration of SFT and RL through the lens of entropy dynamics to clarify their complementary roles in LLM fine-tuning. We begin with analyzing two sequential integration approaches: SFT→RL and RL→SFT, as shown in Figure 4. As demonstrated in Table 1 and Figure 4(a), applying SFT after RL consistently yields suboptimal performance across all benchmarks. To mitigate the policy shifts induced by RL→SFT, we introduce

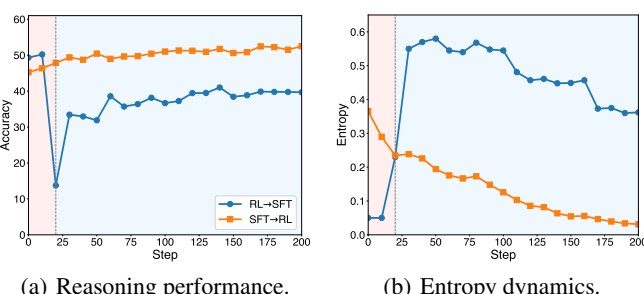

(a) Reasoning performance.  (b) Entropy dynamics.

Figure 4: Comparison between SFT→RL and RL→SFT.

a KL divergence constraint (SFT$_{KL}$, detailed in Appendix G). However, even with this constraint, the performance improvements remained limited, suggesting fundamental incompatibility in this paradigm. In contrast, existing methods successfully achieve substantial performance gains through RL when applied after the base model SFT, as evidenced in Table 1. This different result reveals that the sequence of fine-tuning paradigms critically affects the final model performance, motivating our entropy-based analysis to understand the underlying mechanisms. To understand this difference, we analyze the training dynamics of SFT and RL from **an entropy perspective**. As illustrated in Figure 4(b), policies after RL exhibit significantly lower entropy, approaching deterministic outputs. However, the distribution shift introduced by subsequent SFT causes a rapid increase in entropy, corresponding to the sharp performance drop at step 20 in Figure 4(a), followed by a gradual decline. Moreover, models after RL demonstrate limited capacity for additional learning through SFT, as evidenced by the entropy plateau occurring after approximately 90 training steps in Figure 4(b). In contrast, base models undergoing SFT exhibit a brief initial entropy increase followed by a sustained decrease, ultimately yielding performance improvements. This distinct entropy trajectory suggests that while RL effectively enhances LLM performance, it simultaneously reduces the model's plasticity (its capacity to adapt through subsequent training). These findings establish entropy as a crucial indicator for effective SFT and RL integration. More discussion about the role of entropy in LLM training can be found in Appendix E.

### 3.2.2 Single-Stage Integration Analysis

Building upon the preceding analysis, we show that the SFT→RL paradigm is better suited for LLM reasoning than RL→SFT. Beyond these sequential approaches, we further investigate a single-stage approach that directly unifies both paradigms (SFT+RL), with a combined objective $\mathcal{L}_{\text{SFT+RL}} = \mathcal{L}_{\text{SFT}} + \mathcal{L}_{\text{RL}}$. We conduct preliminary experiments comparing pure RL, sequential SFT→RL with varying SFT steps, and the single-stage SFT+RL, as illustrated in Figure 5. Our empirical results reveal that the single-stage SFT+RL method achieves **superior training efficiency** compared with both the sequential SFT→RL and pure RL. We attribute this behavior to two principal factors: First, the SFT dataset derived from other models' responses may not consistently represent optimal solutions, even when curated from high-quality demonstrations, thereby risking overfitting to suboptimal policies.

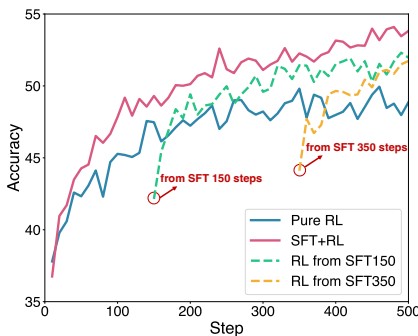

Figure 5: Comparison across pure RL, sequential SFT→RL, and single-stage SFT+RL integration approaches.

Second, pure RL is typically data-inefficient because it cannot exploit existing demonstrations that would guide exploration toward near-optimal solutions. In contrast, the single-stage SFT+RL method effectively leverages demonstrations through unified optimization. This approach enables direct policy optimization toward the target objective while preserving the knowledge distillation benefits of supervised learning from demonstrations.

## 4 Method

In this section, we present the Supervised and Reinforcement Fine-tuning (SRFT), which integrates the advantages of SFT and RL in a single-stage approach. Building upon the RL framework described in Sec. 2, SRFT incorporates flexible guidance from demonstrations, enabling it to harness the strengths of both fine-tuning paradigms. The core innovation of SRFT lies in its single-stage learning mechanism: coarse-grained behavior policy approximation through SFT and fine-grained policy improvement through RL, both applied to demonstration data and self-rollout data.

### 4.1 Learning from Demonstrations

Given a dataset containing demonstrations $\mathcal{D}_{\text{demo.}}$ (e.g., reasoning responses generated by expert LLMs), SRFT employs a dual-pronged strategy to harness this valuable resource. First, we leverage SFT to perform a coarse-grained approximation of the behavior policy underlying the expert's responses. The behavior policy $\pi_\beta(\boldsymbol{y}|\boldsymbol{x})$ captures the underlying generation patterns that produced these high-quality responses, which we seek to approximate through supervised learning: $\mathcal{L}_{\text{SFT}}^{\text{demo.}} = \mathbb{E}_{(\boldsymbol{x},\boldsymbol{y}) \sim \mathcal{D}_{\text{demo.}}}[-\log \pi_\theta(\boldsymbol{y}|\boldsymbol{x})]$. Second, we adopt an off-policy RL approach similar to LUFFY (Yan et al., 2025) to perform fine-grained learning of the behavior policy. Specifically, we directly augment the LLM's rollout data with demonstrations, creating a heterogeneous training group:

$$G_{\text{aug.}} = \{(\boldsymbol{x}_i, \boldsymbol{y}_i)\}_{i=1}^{|G_{\text{roll.}}|} \cup \{(\boldsymbol{x}_j, \boldsymbol{y}_j)\}_{j=1}^{|G_{\text{demo.}}|}, \tag{6}$$

where $G_{\text{roll.}}$ denotes the on-policy rollout group, $G_{\text{demo.}}$ denotes the off-policy demonstration group. The advantage estimation for the entire group is given by:

$$\hat{A}_k = \frac{r(\boldsymbol{x}, \boldsymbol{y}_k) - \text{mean}\{r(\boldsymbol{x}, \boldsymbol{y}_k)|k=1,2,\ldots,|G_{\text{aug.}}|\}}{\text{std}\{r(\boldsymbol{x}, \boldsymbol{y}_k)|k=1,2,\ldots,|G_{\text{aug.}}|\}}. \tag{7}$$

Since responses generated by experts typically yield higher rewards, this augmentation mechanism effectively increases the advantage estimates for the entire group, as shown in Eq. (2), thereby promoting **optimistic exploration** of the LLM policy (Ciosek et al., 2019).

To address the distribution mismatch between behavior policies $\pi_\beta$ of demonstrations and the training policy $\pi_\theta$, we implement two mitigation strategies: **For SFT on demonstrations**, our entropy analysis in Sec. 3.2 demonstrates that entropy serves as a crucial indicator for the effective

integration of SFT and RL. Building on this insight, we introduce an adaptive weighting mechanism that dynamically adjusts based on the current policy entropy. Specifically, we employ $w_{\text{SFT}} = 0.5 \times \texttt{stop\_grad}(\exp(-\mathcal{H}(\pi_\theta)))$ as the SFT weight, where $\texttt{stop\_grad}(\cdot)$ denotes the gradient-stopping operation. This entropy-aware mechanism ensures that when the policy exhibits high entropy (indicating uncertainty), the SFT loss exerts diminished influence on updates, thereby mitigating performance degradation caused by distribution mismatch between policies:

$$\mathcal{L}_{\text{SFT}}^{\text{demo.}}(\theta) = w_{\text{SFT}} \cdot \mathbb{E}_{(\boldsymbol{x},\boldsymbol{y}) \sim \mathcal{D}_{\text{demo.}}}[-\log \pi_\theta(\boldsymbol{y}|\boldsymbol{x})]. \tag{8}$$

**For off-policy RL training**, we introduce an importance sampling ratio similar to GRPO (Shao et al., 2024) and PPO (Schulman et al., 2017) to account for distribution mismatch:

$$\mathcal{L}_{\text{RL}}^{\text{demo.}}(\theta) = -\mathbb{E}_{(\boldsymbol{x},\boldsymbol{y}) \sim \mathcal{D}_{\text{demo.}}}\left[\min\left\{r_{k,t}(\theta) \cdot \hat{A}_k, \texttt{clip}\left(r_{k,t}(\theta), 1-\epsilon, 1+\epsilon\right) \cdot \hat{A}_k\right\}\right], \tag{9}$$

where $r_{k,t}(\theta) = \frac{\pi_\theta(y_{k,t}|(\boldsymbol{x},\boldsymbol{y}_{<t}))}{\pi_\beta(y_{k,t}|(\boldsymbol{x},\boldsymbol{y}_{<t}))}$. Following established practices in recent work (Yan et al., 2025; Ma et al., 2025), we set the behavior policy $\pi_\beta = 1$ to avoid tokenization complexities when aligning the behavior and training policies. This approach enables straightforward integration of off-the-shelf datasets without requiring recomputation of behavior policy probabilities.

## 4.2 LEARNING FROM SELF-EXPLORATION

In addition to leveraging the demonstration data, SRFT enables the LLM policy to learn from its own exploration rollouts. Although traditional RL methods learn from both positive and negative samples generated during rollouts, we observe that under on-policy RL with binary rewards $\{1, -1\}$, the standard RL objective function can be naturally decomposed into two distinct components:

$$\begin{aligned}
\mathcal{L}_{\text{RL}}^{\text{self-rollout}} &= -\mathbb{E}_{\boldsymbol{x} \sim \mathcal{D}, \boldsymbol{y} \sim \pi_\theta(\cdot|\boldsymbol{x})}[R(\boldsymbol{x},\boldsymbol{y})\log \pi_\theta(\boldsymbol{y}|\boldsymbol{x})] \\
&= \underbrace{\mathbb{E}_{\boldsymbol{x} \sim \mathcal{D}, \boldsymbol{y}^+ \sim \pi_\theta(\cdot|\boldsymbol{x})}[-\log \pi_\theta(\boldsymbol{y}^+|\boldsymbol{x})]}_{\text{Positive Sample } \textcircled{1}} + \underbrace{\mathbb{E}_{\boldsymbol{x} \sim \mathcal{D}, \boldsymbol{y}^- \sim \pi_\theta(\cdot|\boldsymbol{x})}[\log \pi_\theta(\boldsymbol{y}^-|\boldsymbol{x})]}_{\text{Negative Sample } \textcircled{2}},
\end{aligned} \tag{10}$$

where $\mathcal{D}$ denotes the RL training dataset, and $\boldsymbol{y}^+$ and $\boldsymbol{y}^-$ represent the correct and incorrect responses, respectively. A critical insight emerges from this decomposition: the positive-sample objective $\textcircled{1}$ is structurally analogous to supervised fine-tuning, as it maximizes the likelihood of correct responses. However, these positive samples are **generated on-policy by the current training policy rather than sourced from SFT datasets**, distinguishing our approach from supervised learning paradigms. The negative-reward component $\textcircled{2}$ implements likelihood minimization, systematically decreasing the probability mass assigned to incorrect responses. This structural correspondence implies that learning from positive samples constitutes a coarse-grained optimization strategy that necessitates careful balance. Moreover, in contrast to learning from demonstrations, self-exploration induces rapid entropy reduction as the model converges toward increasingly deterministic outputs, potentially compromising exploration capabilities. To mitigate this phenomenon and preserve training stability, inspired by our analysis in Sec. 3.1.1, we introduce an entropy-adaptive weighting mechanism $w_{\text{RL}} = 0.1 * \texttt{stop\_grad}(\exp(\mathcal{H}(\pi_\theta)))$ specifically for the positive sample objective. This mechanism parallels our formulation in Eq. (8) but serves the purpose of maintaining the policy's exploration. The complete self-rollout objective is formulated as:

$$\mathcal{L}_{\text{RL}}^{\text{self-rollout}}(\theta) = w_{\text{RL}} \cdot \mathbb{E}_{\boldsymbol{x} \sim \mathcal{D}, \boldsymbol{y}^+ \sim \pi_\theta(\cdot|\boldsymbol{x})}[-\log \pi_\theta(\boldsymbol{y}^+|\boldsymbol{x})] + \mathbb{E}_{\boldsymbol{x} \sim \mathcal{D}, \boldsymbol{y}^- \sim \pi_\theta(\cdot|\boldsymbol{x})}[\log \pi_\theta(\boldsymbol{y}^-|\boldsymbol{x})]. \tag{11}$$

## 4.3 INTEGRATING DEMONSTRATIONS WITH SELF-EXPLORATION ROLLOUTS IN A SINGLE-STAGE APPROACH

By leveraging both demonstrations and self-exploration rollouts, SRFT effectively balances the coarse-grained adjustments of SFT with the fine-grained refinements of RL throughout the single-stage fine-tuning process. The total loss function combines all three components:

$$\mathcal{L}_{\text{SRFT}}(\theta) = \mathcal{L}_{\text{SFT}}^{\text{demo.}}(\theta) + \mathcal{L}_{\text{RL}}^{\text{demo.}}(\theta) + \mathcal{L}_{\text{RL}}^{\text{self-rollout}}(\theta). \tag{12}$$

## 5 EXPERIMENTS

### 5.1 EXPERIMENTAL SETUPS

**Training Datasets.** We employ OpenR1-Math-46k-8192 (Yan et al., 2025) as the training dataset for SRFT, accompanied by high-quality reasoning responses generated by DeepSeek-R1 (Guo et al., 2025). The dataset undergoes filtering through Math-Verify to exclude instances with unverifiable answers or responses exceeding 8,192 tokens in length. This dataset serves multiple purposes in our framework: providing prompts for policy rollouts, ground-truth answers for reward computation, and high-quality demonstrations for SRFT. Additional dataset details are provided in Appendix H.

**Evaluation.** We conduct a comprehensive evaluation on widely-adopted mathematical reasoning benchmarks, including AIME24 (Li et al., 2024), AMC (Li et al., 2024), Minerva (Lewkowycz et al., 2022), OlympiadBench (He et al., 2024), and MATH500 (Hendrycks et al., 2021). For datasets with limited sample sizes (AIME24 and AMC), we report avg@32; for the remaining datasets, we adopt pass@1 as the evaluation criterion. We further assess the model's generalization ability on three out-of-distribution benchmarks: ARC-C (Clark et al., 2018), GPQA-Diamond (Rein et al., 2024) (denoted as GPQA-D), and MMLU-Pro (Wang et al., 2024).

**Baseline Methods.** We benchmark SRFT against the following baselines using Qwen2.5-Math-7B as the base model. **SFT methods:** (1) SFT training on the same dataset; (2) SFT training with KL divergence constraints ($SFT_{KL}$) on the same dataset. **RL methods:** (3) $RL_{GRPO}$ (Shao et al., 2024), a simplified PPO variant trained on the same dataset; (4) Simple-RL-Zero (Zeng et al., 2025), applying GRPO to 24k mathematical samples from GSM8K and MATH; (5) OpenReasoner-Zero (Hu et al., 2025a), a PPO-based approach trained on 129k multi-source samples; (6) PRIME-Zero (Cui et al., 2025a), conducting policy rollouts on 150k NuminaMath queries with implicit process rewards and final labels. **SFT and RL methods:** (7) SFT→RL, sequential training with SFT on the same dataset followed by GRPO; (8) SFT+RL, simultaneous training with SFT and RL on the same dataset; (9) ReLIFT (Ma et al., 2025), an approach that interleaves RL with online Fine-Tuning on the hardest questions; (10) LUFFY (Yan et al., 2025), a mixed-policy GRPO approach using the same dataset; (11) TAPO (Wu et al., 2025), dynamically integrating structured external knowledge within the GRPO framework, trained on 5.5k samples from the MATH dataset.

**Implementation Details.** Building on recent work (Yan et al., 2025; Cui et al., 2025a), we adopt Qwen2.5-Math-7B (Yang et al., 2024) as the base model. In SRFT, we generate eight rollout trajectories per prompt with a maximum sequence length of 8,192 tokens. All experiments are conducted for 500 training steps. The source code is available at here, and the comprehensive experimental details are provided in Appendix G.

### 5.2 EXPERIMENTAL RESULTS

**Reasoning Benchmark Performance.** Our main results are shown in Table 2, where we compare SRFT with zero-RL baselines, as well as direct SFT, SFT→RL, and SFT+RL methods. Across five challenging competition-level reasoning benchmarks, SRFT achieves an average score of **59.1**, significantly outperforming existing RL methods by a margin of **+9.0** points over the best baseline. This clearly demonstrates the advantages of integrating demonstrations with self-exploration in LLM reasoning. We also observe that SRFT achieves a **+4.8** points improvement over SFT methods, indicating that the self-exploration component effectively refines the policy distribution learned from demonstrations. Compared to the SFT→RL and SFT+RL methods, SRFT achieves **+3.4** and **+7.2** points improvement, respectively. These results demonstrate that our single-stage design and entropy-aware weighting mechanism effectively balance the benefits of demonstrations and self-exploration. Additional results are provided in Appendix A.1.

**Out-of-Distribution Generalization.** Regarding out-of-distribution performance, the results in Table 2 show that SRFT also achieves an average score of **62.5** and exceeds the best baseline by **+4.7** points. These results highlight that SRFT effectively combines demonstrations with self-exploration to mitigate the potential overfitting associated with SFT, thereby improving generalization.

**Training Dynamics.** Figure 6 presents the training dynamics of SRFT, including training rewards, response lengths, and training entropy. The results demonstrate that SRFT achieves more rapid improvement than RL, with both approaches exhibiting monotonically increasing training rewards.

Table 2: Overall performance. **Bold** and underlined indicate the best and second-best performance.

| Model | In-Distribution Performance | | | | | | Out-of-Distribution Performance | | | |
|---|---|---|---|---|---|---|---|---|---|---|
| | AIME24 | AMC | MATH500 | Minerva | Olympiad | Avg. | ARC-C | GPQA-D | MMLU-Pro | Avg. |
| **Qwen2.5-Math-7B** | 11.4 | 32.6 | 48.8 | 8.7 | 15.8 | 23.5 | 18.2 | 11.1 | 16.9 | 15.4 |
| **Qwen2.5-Math-7B-Instruct** | 12.9 | 48.3 | 81.2 | 33.1 | 39.8 | 43.1 | 70.3 | 24.7 | 34.1 | 43.0 |
| *Supervised Fine-Tuning* | | | | | | | | | | |
| **SFT** | 31.1 | 62.8 | 85.2 | 39.1 | 53.3 | 54.3 | 76.2 | 25.8 | 45.7 | 49.2 |
| **SFT$_{KL}$** | 13.0 | 45.2 | 70.2 | 26.5 | 36.3 | 38.2 | 33.3 | 22.2 | 30.4 | 28.6 |
| *Reinforcement Learning* | | | | | | | | | | |
| **RL$_{GRPO}$** (Shao et al., 2024) | 24.7 | 61.6 | 79.2 | 33.7 | 47.1 | 49.3 | 75.6 | 31.3 | 42.1 | 49.7 |
| **SimpleRL-Zero**[*] (Zeng et al., 2025) | 27.0 | 54.9 | 76.0 | 25.0 | 34.7 | 43.5 | 30.2 | 23.2 | 34.5 | 29.3 |
| **OpenReasoner-Zero**[*] (Hu et al., 2025a) | 16.5 | 52.1 | 82.4 | 33.1 | 47.1 | 46.2 | 66.2 | 29.8 | **58.7** | 51.6 |
| **PRIME-Zero**[*] (Cui et al., 2025a) | 17.0 | 54.0 | 81.4 | 39.0 | 40.3 | 46.3 | 73.3 | 18.2 | 32.7 | 41.4 |
| **Oat-Zero**[*] (Zeng et al., 2025) | 33.4 | 61.2 | 78.0 | 34.6 | 43.4 | 50.1 | 70.1 | 23.7 | 41.7 | 45.2 |
| *SFT and RL* | | | | | | | | | | |
| **SFT→RL** | 32.5 | 67.1 | 84.2 | 34.1 | 54.6 | 54.5 | 76.4 | 37.9 | 49.6 | 54.6 |
| **SFT+RL** | 26.6 | 69.9 | 83.7 | 35.8 | 51.4 | 52.3 | 74.2 | 34.1 | 44.5 | 50.9 |
| **LUFFY**[*] (Yan et al., 2025) | 29.4 | 65.6 | 87.6 | 37.5 | 57.2 | 55.5 | 80.5 | 39.9 | 53.0 | 57.8 |
| **TAPO**[*] (Wu et al., 2025) | 33.3 | **77.5** | 83.4 | 38.2 | 46.2 | 55.7 | 81.6 | 37.9 | 49.6 | 56.4 |
| **ReLIFT** (Ma et al., 2025) | 28.2 | 64.8 | 85.0 | 37.1 | 54.9 | 54.0 | 74.9 | 40.9 | 51.9 | 55.9 |
| **SRFT (ours)** | **35.3** | 74.3 | **89.8** | **39.7** | **58.3** | **59.5** | **85.3** | **46.4** | 55.9 | **62.5** |

[*]This method's performance is taken from the corresponding paper.

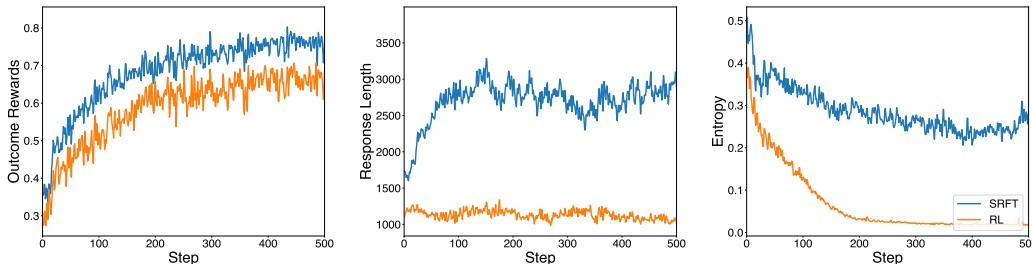

Figure 6: Training dynamics during SRFT and RL training, including training rewards (left), response lengths (middle), and training entropy (right).

This pattern indicates that SRFT benefits from the sample efficiency provided by learning from demonstrations. Regarding response length, when confronted with challenging training data, RL exhibits a tendency toward generating more concise responses, whereas SRFT shows a progressive lengthening of responses, indicating the development of more comprehensive and detailed reasoning processes. From an entropy perspective, SRFT maintains relatively stable entropy throughout training, contrasting with RL's rapid entropy collapse. This stability indicates that the policy maintains exploration during training, validating the effectiveness of the entropy-aware weighting mechanism. The dynamics of the sequence mean for weights $w_{SFT}$ and $w_{RL}$ are visualized in Appendix A.2.

**Ablation Study.** We conduct a comprehensive ablation study to evaluate the impact of the entropy-aware weighting mechanisms and the necessity of each training component. To validate the effectiveness of our adaptive weighting mechanism, we first conduct an ablation study on $w_{SFT}$ and $w_{RL}$. We substitute these entropy-aware dynamic weights with fixed constants, specifically $\overline{w}_{SFT} = 0.1$ and $\overline{w}_{RL} = 1.0$. As detailed in Table 3, employing a fixed SFT weight causes a substantial performance decline to 55.1, while a fixed RL weight reduces accuracy to 56.2. These outcomes corroborate our analysis in Sec. 3, confirming that using entropy as a dynamic indicator significantly outperforms static weighting schemes by effectively balancing exploration and exploitation. Second, we investigate the contribution of specific loss components by selectively removing the SFT loss on demonstrations (w/o demo$_{SFT}$) and the off-policy RL loss on demonstrations (w/o demo$_{RL}$). Note that since SRFT is fundamentally built upon the RL framework, the self-rollout component ($\mathcal{L}_{RL}^{self\text{-}rollout}$) is consistently retained. We observe that eliminating the demonstration SFT yields the most severe degradation, underscoring the critical role of coarse-grained supervised guidance. Similarly, removing

Table 3: Ablation study on weighting mechanisms and loss components. $\overline{w}$ denotes using fixed weights, and w/o denotes removing the specific loss component.

| Method | AIME24 | AMC | MATH500 | Minerva | Olympiad | Avg. |
|---|---|---|---|---|---|---|
| w/ $\overline{w}_{\text{SFT}}$ | 30.1 | 65.8 | 87.0 | 36.8 | 55.8 | 55.1 |
| w/ $\overline{w}_{\text{RL}}$ | 32.6 | 67.2 | 87.5 | 37.4 | 56.6 | 56.2 |
| w/o demo$_{\text{SFT}}$ | 29.5 | 65.2 | 85.6 | 35.9 | 53.1 | 53.9 |
| w/o demo$_{\text{RL}}$ | 30.8 | 68.0 | 86.4 | 36.2 | 54.5 | 55.2 |
| SRFT | 35.3 | 74.3 | 89.8 | 39.7 | 58.3 | 59.5 |

the demonstration RL component lowers performance to 55.2. Collectively, these results demonstrate that all components are essential for SRFT, jointly contributing to the final reasoning capability.

## 6 RELATED WORK

**Reinforcement Learning for LLM Reasoning.** Recent RL approaches (Shao et al., 2024; Guo et al., 2025; Yu et al., 2025; Liu et al., 2025c) have demonstrated substantial improvements in mathematical reasoning. However, the mechanisms through which RL enhances reasoning remain incompletely understood (Yue et al., 2025a; Wang et al., 2025a; Liu et al., 2025d). In this work, we introduce a single-stage method, SRFT, that integrates SFT with RL to maintain stable entropy throughout training while delivering sustained performance improvements.

**Integrating Supervised Fine-Tuning and Reinforcement Learning.** The integration between SFT and RL represents a critical frontier in recent LLM research. While SFT establishes robust policy foundations, RL enables optimization beyond potentially suboptimal SFT trajectories (Cai et al., 2025; Chen et al., 2025a). Recent approaches integrate external supervision into RL frameworks to enhance sample efficiency: UFT (Liu et al., 2025a) merges SFT and RL using partial solution hints; LUFFY (Yan et al., 2025) augments on-policy learning with off-policy traces from stronger models; ReLIFT (Ma et al., 2025) interleaves RL with SFT on high-quality demonstrations; TAPO (Wu et al., 2025) incorporates external "thought patterns" for balanced exploration; SASR (Chen et al., 2025b) provides a theoretical framework unifying SFT and RL with adaptive blending; and CHORD (Zhang et al., 2025a) proposes to adjust the weights of SFT and RL based on the training dynamics. Single-stage integration also mitigates catastrophic forgetting encountered in sequential SFT→RL transitions (Chen et al., 2025b; Liu et al., 2025b). In this work, we first analyze the different roles of SFT and RL in fostering LLM reasoning capabilities. Motivated by this analysis, we introduce SRFT, a single-stage method that dynamically integrates SFT and RL through an entropy-aware weighting mechanism. More detailed background is provided in Appendix F.

## 7 CONCLUSION

In this work, we investigate the integration of SFT and RL for LLM reasoning. Through comprehensive analysis, we reveal that SFT performs coarse-grained global adjustments while RL conducts fine-grained selective optimization, with entropy serving as a crucial indicator of training dynamics. Building on these insights, we propose SRFT, a single-stage approach that unifies both paradigms through entropy-aware weighting mechanisms. Extensive experiments demonstrate the effectiveness of SRFT, which achieves 59.1% average accuracy and outperforms zero-RL baselines by 9.0% on reasoning tasks and 10.9% on OOD benchmarks. Future work could explore extending SRFT to agentic RL scenarios (Zhang et al., 2025b) and investigating its performance with imperfect demonstrations.

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

# Appendix

## A  ADDITIONAL EXPERIMENTAL RESULTS AND ANALYSIS

### A.1  PERFORMANCE COMPARISON ON DIFFERENT MODELS AND BENCHMARKS

We further evaluate the performance of SRFT on Llama-3.1-8B (Grattafiori et al., 2024), Qwen2.5-Math-1.5B and Qwen2.5-7B-Instruct models, as shown in Table A1. We adopt the same training settings as in the main experiment, including training data and hyperparameter configurations. The results show that our method achieves superior performance compared to baseline approaches across most benchmarks. This confirms the effectiveness and scalability of our approach across different model series and sizes.

Table A1: Performance comparison on mathematical reasoning benchmarks on Qwen2.5-Math-1.5B-Base and Qwen2.5-7B-Instruct models. **Bold** indicates the best performance.

| Model | AIME24 | AMC23 | MATH500 | Minerva | Olympiad | Avg. |
|---|---|---|---|---|---|---|
| **Qwen2.5-Math-1.5B** | 7.2 | 26.4 | 28.0 | 9.6 | 21.2 | 18.5 |
| **SFT** | 11.7 | 37.8 | 70.6 | 26.8 | 31.3 | 35.6 |
| **GRPO** | 11.8 | 40.2 | 61.8 | 26.8 | 32.0 | 34.5 |
| **LUFFY** | 14.3 | 43.3 | 74.2 | 26.5 | **39.9** | 39.6 |
| **SRFT** | **15.5** | **44.9** | **74.6** | **28.7** | 39.8 | **40.7** |
| **Qwen2.5-7B-Instruct** | 11.7 | 43.8 | 71.8 | 30.9 | 40.4 | 39.7 |
| **SFT** | 7.9 | 36.0 | 68.6 | 21.3 | 31.1 | 33.0 |
| **GRPO** | 14.1 | 43.5 | 74.0 | 33.8 | 37.6 | 40.6 |
| **LUFFY** | **17.7** | 50.9 | 82.0 | 31.3 | **47.4** | 45.9 |
| **SRFT** | 16.6 | **51.0** | **82.2** | **34.6** | 46.9 | **46.3** |
| **Llama-3.1-8B** | 0.3 | 4.2 | 13.8 | 4.4 | 3.8 | 5.2 |
| **SFT** | 0.5 | 5.4 | 20.2 | 4.0 | 5.3 | 7.1 |
| **GRPO** | 0.3 | 9.4 | 23.4 | **17.6** | 6.1 | 11.4 |
| **LUFFY** | 1.9 | 13.5 | 39.0 | 15.1 | **9.6** | 15.8 |
| **SRFT** | **1.9** | **14.3** | **40.1** | 15.3 | 9.5 | **16.2** |

Besides, we further compare the performance of SRFT with other baselines on the AIME25 dataset (Li et al., 2024), as shown in Table A2. We observe that SRFT achieves superior performance compared to baselines, outperforming SFT by +1.3 points and GRPO by +7.1 points. This validates the effectiveness of our method in simultaneously leveraging the advantages of SFT and RL within a single-stage framework for reasoning tasks.

Table A2: Performance Comparison on AIME25.

| Model | AIME25 |
|---|---|
| **Qwen2.5-Math-7B** | 4.9 |
| **Qwen2.5-Math-7B-Instruct** | 10.1 |
| **SFT** | 20.3 |
| **GRPO** | 14.5 |
| **SFT→RL** | 21.2 |
| **SFT+RL** | 15.6 |
| **SRFT** | 21.6 |

Table A3: Performance comparison under the evaluation setting proposed by Hochlehnert et al. (2025). We report the mean accuracy and standard deviation across tasks.

| Model | AIME24 | AMC | MATH-500 | Minerva | Olympiad | Avg. |
|---|---|---|---|---|---|---|
| Qwen2.5-Math-7B | $20.7 \pm 3.8$ | $56.2 \pm 5.7$ | $64.3 \pm 0.5$ | $17.3 \pm 1.9$ | $29.0 \pm 0.5$ | 37.5 |
| $RL_{GRPO}$ | $22.5 \pm 2.6$ | $64.4 \pm 3.4$ | $78.6 \pm 0.3$ | $32.8 \pm 1.3$ | $42.3 \pm 0.9$ | 48.1 |
| SRFT (Ours) | $28.3 \pm 2.4$ | $71.8 \pm 3.9$ | $88.0 \pm 0.2$ | $34.4 \pm 1.2$ | $54.7 \pm 0.5$ | 55.5 |

Following the evaluation protocols proposed in recent work Hochlehnert et al. (2025), we present additional results in Table A3. The results demonstrate that SRFT maintains a performance edge, outperforming the base model by an average of 18.0 points and $RL_{GRPO}$ by 7.4 points. These findings corroborate the robustness of our method, confirming that the reported gains are consistent across varying conditions and not merely artifacts of a specific evaluation setting.

## A.2 DYNAMICS OF THE ENTROPY-AWARE WEIGHTS

In order to better understand the dynamics of entropy-aware weights, we visualize the temporal evolution of the sequence mean for $w_{\text{SFT}}$ and $w_{\text{RL}}$ throughout the training process, as illustrated in Figure A1. It is important to clarify that $w_{\text{SFT}}$ and $w_{\text{RL}}$ serve distinct mechanisms: $w_{\text{SFT}}$ balances the SFT and off-policy RL objectives specifically for demonstration data, whereas $w_{\text{RL}}$ balances the positive and negative sample objectives for rollout data. Consequently, their relative magnitudes do not represent the global loss contribution ratio between demonstration learning and self-exploration, which are weighted equally in the total objective (Eq. (12)). Regarding their temporal behaviors, we observe that $w_{\text{SFT}}$ gradually increases to $0.40$ as training progresses, whereas $w_{\text{RL}}$ fluctuates around $0.13$. These observations demonstrate that our method can effectively adjust the internal balance of objectives in response to training dynamics, thereby ensuring consistent exploration throughout the optimization process. Besides, the baseline coefficients of $0.5$ and $0.1$ in the weight formulations are hyperparameters that can be fine-tuned to further improve performance.

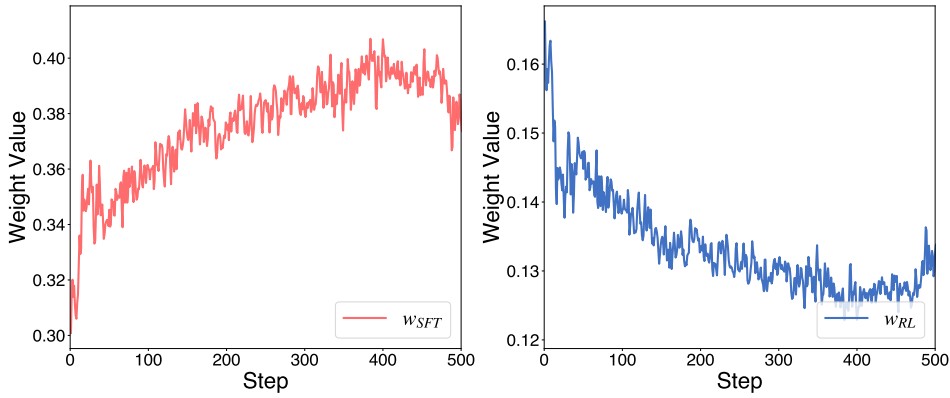

Figure A1: Dynamics of the sequence mean for $w_{\text{SFT}}$ and $w_{\text{RL}}$ during SRFT training.

## A.3 ANALYSIS OF ANNEALING DEMONSTRATION WEIGHTS

To investigate the influence of SFT on the RL process in SRFT, we designed an experiment where the weight of the demonstration data is annealed over time. We modify the total objective function by introducing a time-dependent coefficient $\alpha$, formulated as:

$$\mathcal{L}_{\text{SRFT}}(\theta) = \alpha \left( \mathcal{L}_{\text{SFT}}^{\text{demo.}}(\theta) + \mathcal{L}_{\text{RL}}^{\text{demo.}}(\theta) \right) + \mathcal{L}_{\text{RL}}^{\text{self-rollout}}(\theta), \tag{A1}$$

where $\alpha$ decays linearly from 1 to 0 over the initial 150 and 300 training steps, respectively. The results are summarized in Table A4, indicating that annealing the demonstration weights—at either 150 or 300 steps—leads to a performance degradation compared to the standard SRFT approach. We attribute this to the high quality of the available demonstrations; the underlying behavior policy is

Table A4: Performance comparison of annealing the demonstration learning component $\alpha$ at different step counts.

| Annealing Steps | AIME24 | AMC | MATH-500 | Minerva | Olympiad | Avg. |
|---|---|---|---|---|---|---|
| 150 | 31.4 | 69.5 | 86.2 | 36.1 | 53.8 | 55.4 |
| 300 | 33.8 | 74.6 | 88.4 | 38.2 | 56.4 | 58.3 |
| w/o annealing | 35.3 | 74.3 | 89.8 | 39.7 | 58.3 | 59.5 |

already near-optimal (DeepSeek-R1), maintaining the supervised signal throughout the RL process proves beneficial for stability and guidance. However, we hypothesize that over significantly extended training horizons, persistent reliance on demonstrations might eventually constrain exploration and impede RL performance. Investigating the dynamics of SFT over long-term RL training (Liu et al., 2025d; Hu et al., 2025b; Khatri et al., 2025) remains a promising avenue for future research.

## A.4 DYNAMICS OF SFT-INITIALIZED RL AND SRFT

To investigate the impact of warm-start initialization, we further analyze the length and reward curves of the sequential SFT→RL approach, as shown in Figure A2. We observe that when RL is initialized with SFT, the response length maintains approximately 4k tokens, exceeding the ~3k length of SRFT. However, despite the increased length, the performance of SFT→RL exhibits premature convergence. In contrast, SRFT sustains continuous performance improvement driven by its unified, single-stage learning from both demonstration and rollout data. This comparison underscores the superiority of SRFT in effectively leveraging demonstrations to guide exploration without suffering from the stagnation observed in sequential paradigms.

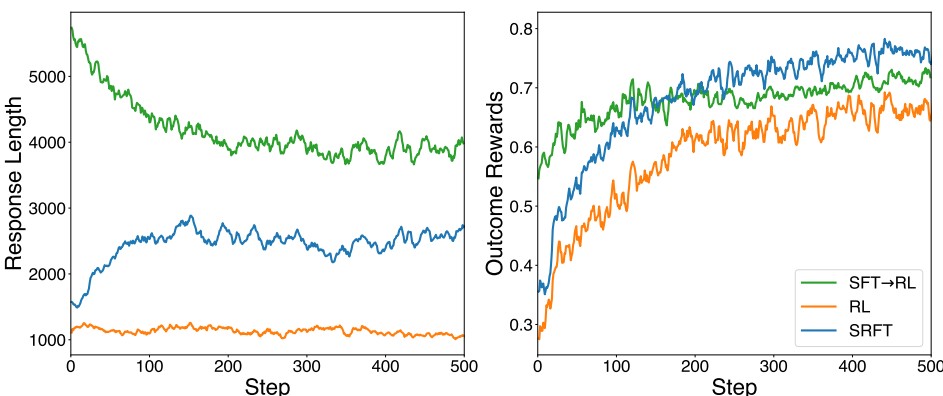

Figure A2: Response lengths and training rewards of pure RL, SFT→RL, and SRFT.

## B    Token Distribution Analysis

In this section, we provide the detailed experimental setting and mathematical derivation for the gradient of the Supervised Fine-Tuning (SFT) loss function, which was referenced in Sec. 3.1.1.

### B.1    Experimental Setting for Visualization

In Sec. 3.1.1, we visualize token distribution changes during SFT and RL training. We use Qwen2.5-Math-7B as the base model, with training data from the main experiment and visualization prompts from the five evaluation benchmarks.

**Methodology.** The response sequences in Figure 2(a) are generated independently by the SFT and RL models from the same prompt. We exclude the base model's outputs from generation, as they are frequently incorrect or too concise to support effective analysis. This approach allows us to observe how fine-tuning redistributes probability mass along the model's own generation trajectory. For each token $y_t$ in a sequence $\boldsymbol{y} = (y_1, \ldots, y_T)$, we compute two conditional probabilities: $p_{\text{fine-tuned}}(y_t|\boldsymbol{y}_{<t})$ from the fine-tuned model (SFT or RL) and $p_{\text{base}}(y_t|\boldsymbol{y}_{<t})$ from the base model. Although the sequence is generated by the fine-tuned model, probabilities are queried from both models using the identical context.

**Visualization Scheme.** To quantify the distributional impact of fine-tuning, we calculate the probability shift: $\Delta_p = p_{\text{fine-tuned}}(y_t|\boldsymbol{y}_{<t}) - p_{\text{base}}(y_t|\boldsymbol{y}_{<t})$, where $\boldsymbol{y}_{<t}$ is the context sequence up to token $y_t$. A diverging color map visually encodes these dynamics:

- **Red** (*Amplification*, $\Delta_p > 0$): Indicates that the fine-tuning process has increased the token's probability relative to the base model.
- **Blue** (*Suppression*, $\Delta_p < 0$): Indicates that the fine-tuning process has decreased the token's probability relative to the base model.
- **Grey/Colorless** (*Retention*, $\Delta_p \approx 0$): Indicates negligible impact, suggesting the fine-tuning largely preserves the base distribution for this token.

This visualization framework facilitates a dual-level analysis: it reveals global trends in probability mass migration at a macro level, while simultaneously highlighting specific tokens that undergo significant distributional shifts at a micro level.

### B.2    SFT Gradient Derivation

**Problem Setup.** Given a demonstration dataset $\mathcal{D} = \{(\boldsymbol{x}_i, \boldsymbol{y}_i)\}_{i=1}^N$ where $\boldsymbol{x}_i$ is an input prompt and $\boldsymbol{y}_i = (y_{i,1}, y_{i,2}, \ldots, y_{i,T_i})$ is the corresponding target sequence, where $T_i = |\boldsymbol{y}_i|$ is the length of the target sequence. The SFT objective function is (the same as Eq. (1)):

$$\mathcal{L}_{\text{SFT}}(\theta) = \mathbb{E}_{(\boldsymbol{x},\boldsymbol{y})\sim\mathcal{D}}[-\log \pi_\theta(\boldsymbol{y}|\boldsymbol{x})]. \tag{A2}$$

Since the sequence probability is factorized as:

$$\pi_\theta(\boldsymbol{y}|\boldsymbol{x}) = \prod_{t=1}^T \pi_\theta(y_t|\boldsymbol{x}, \boldsymbol{y}_{<t}). \tag{A3}$$

The SFT loss becomes:

$$\mathcal{L}_{\text{SFT}}(\theta) = \mathbb{E}_{(\boldsymbol{x},\boldsymbol{y})\sim\mathcal{D}}\left[-\sum_{t=1}^T \log \pi_\theta(y_t|\boldsymbol{x}, \boldsymbol{y}_{<t})\right]. \tag{A4}$$

**Gradient Derivation.** To derive the gradient $\nabla_\theta \mathcal{L}_{\text{SFT}}(\theta)$, we need to compute:

$$\nabla_\theta \mathcal{L}_{\text{SFT}}(\theta) = \mathbb{E}_{(\boldsymbol{x},\boldsymbol{y})\sim D_{\text{data}}}\left[-\sum_{t=1}^T \nabla_\theta \log \pi_\theta(y_t|\boldsymbol{x}, \boldsymbol{y}_{<t})\right]. \tag{A5}$$

Now, let's consider the key insight: at each time step $t$, the model produces a probability distribution over the entire vocabulary $\mathcal{V}$. The gradient of the log probability for the target token $y_t$ can be expressed in terms of the model's prediction probabilities for all vocabulary tokens.

For any token $v \in \mathcal{V}$ at position $t$, we have:

$$\nabla_\theta \log \pi_\theta(v|\boldsymbol{x}, \boldsymbol{y}_{<t}) = \frac{1}{\pi_\theta(v|\boldsymbol{x}, \boldsymbol{y}_{<t})} \nabla_\theta \pi_\theta(v|\boldsymbol{x}, \boldsymbol{y}_{<t}). \tag{A6}$$

Since $\sum_{v \in \mathcal{V}} \pi_\theta(v|\boldsymbol{x}, \boldsymbol{y}_{<t}) = 1$, we have the constraint:

$$\sum_{v \in \mathcal{V}} \nabla_\theta \pi_\theta(v|\boldsymbol{x}, \boldsymbol{y}_{<t}) = 0. \tag{A7}$$

Using the chain rule and the fact that the softmax normalization affects all vocabulary tokens, the gradient can be written as:

$$\nabla_\theta \log \pi_\theta(y_t|\boldsymbol{x}, \boldsymbol{y}_{<t}) = \sum_{v \in \mathcal{V}} \left(\mathbf{1}_{v=y_t} - \pi_\theta(v|\boldsymbol{x}, \boldsymbol{y}_{<t})\right) \nabla_\theta \log \pi_\theta(v|\boldsymbol{x}, \boldsymbol{y}_{<t}), \tag{A8}$$

where $\mathbf{1}_{v=y_t}$ is the indicator function that equals 1 when $v = y_t$ and 0 otherwise.

Substituting this back into the SFT gradient, we obtain:

$$\nabla_\theta \mathcal{L}_{\text{SFT}} = \mathbb{E}_{(\boldsymbol{x}, \boldsymbol{y}) \sim \mathcal{D}} \left[ \sum_{t=1}^{|\boldsymbol{y}|} \sum_{v \in \mathcal{V}} \left(\pi_\theta(v|\boldsymbol{x}, \boldsymbol{y}_{<t}) - \mathbf{1}_{v=y_t}\right) \nabla_\theta \log \pi_\theta(v|\boldsymbol{x}, \boldsymbol{y}_{<t}) \right], \tag{A9}$$

where $\mathcal{V}$ is the LLM vocabulary, and $\mathbf{1}_{v=y_t}$ is an indicator function that equals 1 when token $v$ matches the target token $y_t$ and 0 otherwise. This formulation reveals the fundamental mechanism of SFT: at each time step, the gradient encourages the model to increase the probability of the target token ($\mathbf{1}_{v=y_t} = 1$) while decreasing the probabilities of all other tokens in the vocabulary ($\mathbf{1}_{v=y_t} = 0$). The magnitude of the decrease for each non-target token $v$ is proportional to its current probability $\pi_\theta(v|\boldsymbol{x}, \boldsymbol{y}_{<t})$.

This analysis confirms our empirical observations that SFT produces broad, coarse-grained changes to the model's probability distributions across the entire vocabulary, systematically sharpening the distribution toward the target tokens in the training data.

## C  DETAILS OF VISUALIZATION OF LEARNING DYNAMICS

To characterize model dynamics during training, as introduced in Sec. 3.1.2, we formulate it as a mapping from input prompts to output probability distributions over the vocabulary. Under this view, two models are considered equivalent if they produce identical output distributions for all possible prompts. This perspective allows us to represent each model as a point in a high-dimensional function space, where each point corresponds to a unique conditional probability distribution.

### C.1  THEORETICAL DEFINITION

**The Model Probability Space Formalization.** To visualize the learning dynamics, we conceptualize each model checkpoint as a point in an infinite-dimensional probability space. A model's position is defined by its conditional probability distribution, $\pi(\cdot|\boldsymbol{x})$, over the entire vocabulary for any given prompt $\boldsymbol{x}$. To measure the distance or dissimilarity between a fine-tuned model $\pi_\theta$ and a reference model $\pi_R$, we use the Kullback-Leibler (KL) divergence. The KL divergence $\mathcal{D}_{\text{KL}}(\pi_R \| \pi_\theta)$ quantifies how much the distribution $\pi_\theta$ diverges from the reference distribution $\pi_R$. However, computing the exact KL divergence is computationally intractable for large-scale language models. Therefore, we use cross-entropy as a practical and effective proxy:

$$\begin{aligned} \mathcal{D}_{\text{KL}}(\pi_R \| \pi_\theta) &= H(\pi_R, \pi_\theta) - H(\pi_R) \\ &= H(\pi_R, \pi_\theta) - c, \end{aligned} \tag{A10}$$

where $H(\pi_R)$ denotes the entropy, a constant term $c$ for each reference model. Thus, we can use the cross-entropy to measure the distance between the fine-tuned model and the reference model.

**Measure Learning Dynamics via Reference Models.** To make this high-dimensional space tractable for analysis, we project the model's learning trajectory onto a lower-dimensional space defined by

a set of diverse, high-performing reference models, $\mathcal{R} = \{\pi_{R_1}, \pi_{R_2}, \ldots, \pi_{R_k}\}$. As mentioned in the main text, our reference set includes Qwen2.5-Math-7B, DeepSeek-R1, and QwQ-32B. For any given fine-tuned model checkpoint $\pi_\theta$, its position in this reference-based space is represented by a vector of cross-entropy values:

$$\boldsymbol{d}_{\mathcal{R}}(\pi_\theta) = (d_{R_1}(\pi_\theta), d_{R_2}(\pi_\theta), \ldots, d_{R_k}(\pi_\theta))^{\mathsf{T}}, \tag{A11}$$

where each component of this vector, $d_{R_i}(\pi_\theta) = H(\pi_{R_i}, \pi_\theta)$, measures the divergence of the fine-tuned model from the corresponding reference model $\pi_{R_i}$. **By computing this vector at various stages of training, we can trace the evolution of $\pi_\theta$ as a trajectory** through this constructed space. This process allows us to visualize and compare the optimization paths taken by different fine-tuning methods, such as SFT and RL, revealing their distinct impacts on the model's underlying probability distribution. For visualization in a 3D plot (as in Figure 3), we can select the three most informative reference models as axes or use dimensionality reduction techniques.

## C.2 EXPERIMENTAL SETUP

To instantiate the above definition, we construct a dataset comprising 1,024 response sequences generated by each of the reference models under identical prompts. These prompts are drawn from a diverse mixture of mathematical reasoning benchmarks (including AIME24, Minerva, Olympiad, AMC, and MATH500) to ensure broad coverage across problem domains and difficulty levels.

For each response, we compute the distance between the probabilities assigned by model $\pi_\theta$ to the reference tokens, and then aggregate overall responses to obtain the final distance $d_{R_i}(\pi_\theta)$ for each $R_i$. These distances collectively define the model's position in the projected subspace. We select three reference models to construct the projection basis: (1) DeepSeek-R1 (Guo et al., 2025), representing state-of-the-art reasoning performance; (2) QwQ-32B (Team, 2025), serving as a high-performing but structurally distinct baseline; and (3) Qwen-2.5-Math-7B (Yang et al., 2024), which acts as the base model prior to fine-tuning. Together, they span a spectrum from foundational to advanced capabilities, forming a semantically meaningful coordinate system.

We train the LLM model on the OpenR1-Math-220k dataset, maintaining consistent training settings across all experiments. We construct evaluation datasets using 1,024 prompts sampled from five evaluation benchmark tasks. For every model checkpoint during training, we compute its distance to reference models based on the probability assignments over reference responses. This procedure yields a trajectory in model space that traces the model's evolution throughout training. By comparing these trajectories across training paradigms (e.g., SFT and RL), we reveal distinct optimization dynamics and convergence behaviors. Notably, we observe that during the SFT→RL training pipeline, **the optimization direction in the RL phase opposes that of the SFT phase, imposing an additional learning tax on sequential LLM training.** This phenomenon aligns with findings from recent studies (Shenfeld et al., 2025; Jin et al., 2025a), providing insights into optimizing LLM training efficiency and the integration of SFT and RL strategies.

The three-dimensional distance framework we proposed provides both **theoretical grounding** and **practical interpretability** for analyzing LLM training dynamics. It enables direct comparison of different model variants and reveals how specific training strategies influence progression through the reasoning capability landscape.

## C.3 Learning Dynamics Visualization with Different Epochs SFT

In Sec. 3.1.2, we hypothesize that "SFT *may* overfit to demonstrations, and the subsequent RL attempts to correct this deviation, thereby increasing the learning tax of the two-stage paradigm." To provide a granular analysis of this phenomenon, we investigate the impact of SFT training duration in Figure A3. We observe that this learning tax is sensitive to the number of SFT epochs: extended supervised training (e.g., 2 or 3 epochs) tends to confine the model within the demonstration distribution, thereby increasing the overfitting issue. Conversely, early stopping (e.g., 1 epoch) followed by RL can mitigate this distributional shift. These findings underscore that identifying the optimal SFT epoch is a complex and often underexplored hyperparameter tuning challenge (Chu et al., 2025; Jin et al., 2025b). Our method SRFT addresses this dilemma by integrating SFT and RL into a single stage with adaptive weighting, enabling for direct policy optimization without the need for manual stage-switching.

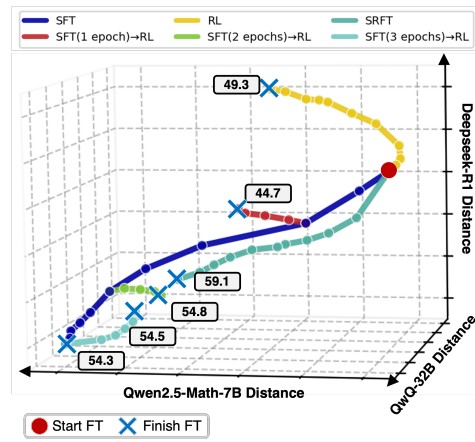

Figure A3: Visualization of learning dynamics across different SFT epochs.

## D Difference between SRFT and On-policy Distillation

In this section, we clarify that a direct comparison between SRFT and on-policy distillation methods such as Generalized Knowledge Distillation (GKD) (Agarwal et al., 2024) is not straightforward due to fundamental theoretical and technical divergences.

**Theoretical Differences.** Fundamentally, SRFT and GKD operate under different paradigms:

- **RL vs. KD Frameworks:** SRFT is grounded in the Reinforcement Learning (RL) framework. Its core innovation lies in the single-stage unification of SFT and RL to achieve direct policy optimization via reward maximization. In contrast, GKD operates within the Knowledge Distillation (KD) framework. As noted by Agarwal et al. (2024), GKD is not an RL algorithm; rather, it aims to mitigate the "distribution mismatch between output sequences seen during training and those generated by the student during inference" by minimizing the divergence between student and teacher distributions.

- **Performance Upper Bounds:** The optimization objective of GKD is to align the student's policy with that of the teacher, which theoretically upper-bounds the student's performance by the teacher's capabilities. Conversely, SRFT utilizes RL to explore the solution space. By leveraging reward signals, SRFT can potentially discover better solution paths and optimize the policy to surpass the performance of the behavior policy.

**Technical Differences.** Implementing on-policy distillation with large-scale reasoning models (e.g., DeepSeek-R1) presents specific technical challenges that SRFT avoids:

- **Vocabulary Mismatch:** Standard GKD requires the teacher and student models to share the same tokenizer vocabulary to compute the token-level KL divergence. In our setting, the teacher (DeepSeek-R1) and the student (Qwen-2.5-Math-7B) utilize different tokenizers. Applying GKD would necessitate complex sequence and vocabulary alignment techniques. SRFT, relying on offline demonstration data, bypassing the need for the teacher's tokenizer.

- **Initialization Requirements:** GKD typically necessitates initialization from an SFT model that can already "generate sequences of adequate quality" for the teacher to provide meaningful feedback. In contrast, our experiments demonstrate that SRFT can effectively optimize the policy directly from the base model.

# E  DISCUSSION ABOUT ENTROPY IN LLM FINE-TUNING FOR REASONING

Entropy is a crucial metric in the LLM reinforcement fine-tuning (Cui et al., 2025b; Cheng et al., 2025; Wang et al., 2025b). It is used to measure the uncertainty of the LLM policy's predictions. The definition of policy entropy is as follows:

$$\mathcal{H}(\pi_\theta) = -\frac{1}{|\boldsymbol{y}|} \sum_{t=1}^{|\boldsymbol{y}|} \sum_{v \in \mathcal{V}} \pi_\theta(v|\boldsymbol{x}, \boldsymbol{y}_{<t}) \log \pi_\theta(v|\boldsymbol{x}, \boldsymbol{y}_{<t}), \tag{A12}$$

where $\pi_\theta$ is the LLM policy, $\mathcal{V}$ is the LLM vocabulary, $\boldsymbol{x}$ is the input prompt, $\boldsymbol{y}_{<t}$ is the sequence of tokens in the response before the current token, and $|\boldsymbol{y}|$ is the length of the response. A higher entropy indicates a more uniform distribution of probabilities across the vocabulary, while a lower entropy indicates a more concentrated distribution.

**Entropy-aware Gradient Clipping.** In this work, we use the entropy of the LLM policy to adjust the weights of the SFT and RL components. Besides, our investigation into the entropy characters of tokens modified by fine-tuning reveals that reinforcement learning predominantly targets tokens with high entropy distributions, a finding that aligns with recent work on selective optimization in language models (Wang et al., 2025c). To empirically validate this observation, we design controlled experiments implementing gradient truncation for high-probability tokens during RL training. As demonstrated in Figure A4, the model's performance remains comparable to the original RL algorithm even when gradients are truncated for low-entropy tokens, providing strong empirical support for our hypothesis. This evidence confirms that RL operates with remarkable selectivity, precisely adjusting tokens with uncertain distributions while leaving confident predictions largely unchanged. In contrast, SFT applies broad modifications across the entire token space, fundamentally altering the model's distributional characters in a less discriminative manner.

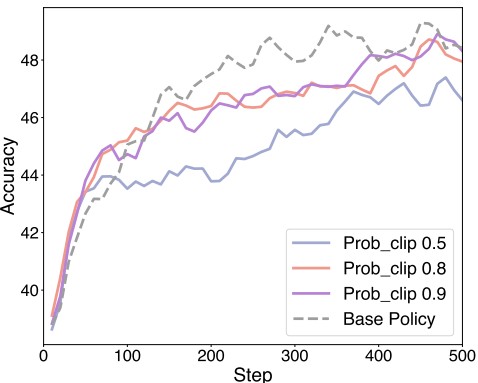

Figure A4: Performance of RL with gradient clipping for low-entropy tokens.

## F BACKGROUND

### F.1 REINFORCEMENT LEARNING

In RL training, the LLM's token generation process is modeled as a Markov Decision Process (MDP) (Puterman, 2014). We define a state $s_t$ at step $t$ as the concatenation of the input prompt $\boldsymbol{x}$ and all tokens generated so far $\boldsymbol{y}_{<t}$. This state serves as input to the policy model $\pi_\theta(\cdot|s_t)$. Specifically, the policy processes $s_t = (\boldsymbol{x}, \boldsymbol{y}_{<t}) = (x_1, x_2, \ldots, x_l, y_1, y_2, \ldots, y_{t-1})$, where $x_i$ denotes the $i$-th token of the input $\boldsymbol{x}$ and $y_j$ represents the token generated by $\pi_\theta$ at step $j$. An action $a_t$ corresponds to the selection of the next output token $y_t$. The LLM, acting as a policy $\pi_\theta(a_t|s_t)$, generates a trajectory $\boldsymbol{y}$ (a sequence of tokens) in response to the prompt $\boldsymbol{x}$. A reward function $R(\boldsymbol{x}, \boldsymbol{y}) = \sum_{t=1}^T r(\boldsymbol{x}, y_t)$ provides a scalar score for the entire trajectory $\boldsymbol{y}$ given prompt $\boldsymbol{x}$, typically derived from human evaluations or automated metrics. In the context of RL, the behavior policy $\pi_\beta(\boldsymbol{y}|\boldsymbol{x})$ refers to the model that generated the responses in the replay buffer. This policy is crucial for RL, particularly for off-policy learning, as it enables proper importance sampling corrections to account for the distribution shift between the data-generating model and the current training model. The MDP formulation in LLMs presents several notable characteristics:

- **Sequential state representation:** At each step $t$, the state $s_t \in S$ consists of the concatenation of the input prompt $\boldsymbol{x}$ and all actions (tokens) generated so far $\boldsymbol{y}_{<t}$. This state serves as input to the policy model $\pi_\theta(\cdot|s_t)$.

- **Sparse and delayed rewards:** Rewards $R(\boldsymbol{x}, \boldsymbol{y})$ are typically sparse, provided only upon completion of a sequence $\boldsymbol{y}$. This dependency on the final output's overall quality complicates credit assignment across the generation process.

### F.2 REINFORCEMENT LEARNING FOR LLM REASONING

The pursuit of complex reasoning capabilities in LLMs has witnessed remarkable progress, with RL emerging as a pivotal methodology for enhancing reasoning abilities (Jiang et al., 2025; Xu et al., 2025) and tool-use capabilities (Chai et al., 2025a; Fu et al., 2025; Chai et al., 2025b; Tu et al., 2025) beyond the limitations of SFT alone. Recent approaches such as GRPO (Shao et al., 2024; Guo et al., 2025), DAPO (Yu et al., 2025), DR.GRPO (Liu et al., 2025c), and VAPO (Yue et al., 2025b) have demonstrated substantial improvements in mathematical reasoning and complex problem-solving tasks. However, the precise mechanisms through which RL enhances reasoning capabilities remain incompletely understood. Several empirical investigations suggest that reinforcement learning primarily serves to elicit, refine, or improve the sampling of pre-existing reasoning abilities rather than instilling entirely novel fundamental reasoning skills from scratch. For instance, Yue et al. (2025a) question whether current reinforcement learning via verifier feedback (RLVF) genuinely expands the reasoning boundary (pass@k) or primarily improves the sampling efficiency of already known solutions (pass@1). Similarly, Wang et al. (2025a) highlight that base models already possess substantial reasoning capabilities that reinforcement learning can effectively unlock or redirect. Nevertheless, ProRL (Liu et al., 2025d) demonstrates that RL-trained models can achieve improved success rates on tasks where base models completely fail, suggesting that sustained and stable reinforcement learning training can indeed expand the reasoning capability boundaries of LLMs. In this work, we design a single-stage method that combines SFT and RL, maintaining stable entropy during training and achieving continuous performance improvement.

### F.3 INTEGRATING SUPERVISED FINE-TUNING AND REINFORCEMENT LEARNING

The synergistic interaction between SFT and RL represents a critical area of investigation in modern LLM development. SFT on high-quality reasoning chains can establish a robust initial policy foundation, which RL can subsequently optimize. Cai et al. (2025) explore the necessary extent of exploration following SFT, finding that RL continues to provide substantial benefits by enabling models to deviate from potentially suboptimal SFT trajectories. Recent research suggests that SFT may equip models with structured reasoning templates that RL subsequently validates and improves (Chen et al., 2025a). Nevertheless, determining the optimal strategy for combining these complementary paradigms remains an active area of debate. To enhance sample efficiency and provide structured guidance for RL exploration, researchers have investigated various approaches

for integrating external supervision into the reinforcement learning framework. UFT (Liu et al., 2025a) proposes a novel paradigm that merges SFT and RL into a single process, using informative supervision signals like hints from partial solutions to guide exploration and accelerate convergence. Addressing the limitations of on-policy learning, LUFFY (Yan et al., 2025) augments RLVR by incorporating off-policy reasoning traces from stronger models, dynamically balancing imitation with on-policy exploration to improve capabilities. ReLIFT (Ma et al., 2025) addresses the limitations of pure RL by interleaving reinforcement learning with supervised fine-tuning on high-quality demonstrations collected during training, enabling models to acquire new knowledge beyond their original capabilities. TAPO (Wu et al., 2025) enhances RL by incorporating external high-level guidance in the form of "thought patterns" abstracted from prior samples, adaptively integrating these to balance model-internal exploration with external strategy exploitation. SASR (Chen et al., 2025b) offers a hybrid framework that theoretically unifies SFT and RL, using SFT for initial warm-up and then adaptively blending it with an online RL method based on training dynamics to maintain core reasoning while exploring diverse paths, using high-quality SFT demonstrations as a key external data source. CHORD (Zhang et al., 2025a) proposes a dynamic weighting scheme that dynamically adjusts the weights of the two components based on the training dynamics, achieving superior performance on both SFT and RL tasks on the instruct model. Furthermore, the single-stage integration of SFT and RL helps mitigate the catastrophic forgetting problem that previous methods encountered when transitioning from SFT to RL (Chen et al., 2025b; Liu et al., 2025b). These approaches collectively underscore an emerging trend toward more sophisticated integrations of supervised signals within reinforcement learning frameworks to improve reasoning alignment and overall performance.

## G   EXPERIMENTAL DETAILS

**Training.** For SFT, we follow the SFT configuration of OpenR1-Qwen-7B (Face, 2025), performing full fine-tuning on DeepSeek-R1 generated reasoning traces and prompts. The training hyperparameters include a batch size of 128, a learning rate of $5 \times 10^{-6}$, a linear learning rate schedule with $10\%$ warmup, and training for 3 epochs. For SFT with KL regularization, we use identical settings while adding a KL divergence regularization between the current policy and the base model (Qwen2.5-Math-7B) with weight $\lambda = 0.2$. The $\text{SFT}_{\text{KL}}$ loss is:

$$\mathcal{L}_{\text{SFT}_{\text{KL}}}(\theta) = \mathbb{E}_{(\boldsymbol{x},\boldsymbol{y})\sim\mathcal{D}}[-\log \pi_\theta(\boldsymbol{y}|\boldsymbol{x})] + \lambda\mathcal{L}_{\text{KL}}(\theta, \theta_{\text{base}}), \tag{A13}$$

where $\mathcal{L}_{\text{KL}}(\theta, \theta_{\text{base}})$ is the KL divergence between the current policy and the base model. For RL, we train for 500 steps with 8 rollouts per prompt. The learning rate is fixed at $1 \times 10^{-6}$. For our method, SRFT, we set the entropy loss coefficient to $0.001$ and remove the KL loss term by setting $\lambda = 0$. To further maintain entropy stability, we adopt the policy shaping mechanism proposed by Yan et al. (2025) and set the shaping value to $\gamma = 0.1$. Additionally, we omit the clipping operation, as the standard clipping mechanism becomes imbalanced and potentially unstable when setting the behavior policy $\pi_\beta = 1$. Since the maximum sequence length for Qwen2.5-Math-7B is 4,096, which is insufficient for our tasks, we increase the RoPE theta from 10,000 to 40,000 and expand the window size to 16,384. For all experiments, we use verl[1] (Sheng et al., 2024) as the implementation framework. We tune hyperparameters for all baselines to ensure fair and optimal performance comparisons. All experiments are conducted on 8x8 A100 GPUs.

**Evaluation.** All evaluations are conducted using vLLM (Kwon et al., 2023) with the sampling temperature set to 0.6 and a maximum generation length of 8,192 tokens. Given the substantial computational costs associated with evaluating multiple RL baselines, we do not perform an exhaustive, model-specific hyperparameter grid-search during evaluation as suggested by Hochlehnert et al. (2025). Instead, to ensure a fair and rigorous comparison, we maintain the same evaluation protocol across all methods, utilizing a unified prompt template, an identical reward function, and consistent inference parameters. Regarding metrics, for datasets with limited sample sizes (AIME24 and AMC), we report the avg@32 metric to reduce variance; for the remaining three datasets, we adopt pass@1 as the primary evaluation criterion. We verify the correctness of generated solutions using Math-Verify[2] and OAT-Grader (Liu et al., 2024). For baseline comparisons, we independently validate the results of the base model, SFT-related baselines, GRPO (Shao et al., 2024), LUFFY (Yan

---

[1] https://github.com/volcengine/verl
[2] https://github.com/huggingface/Math-Verify

et al., 2025), and ReLIFT (Ma et al., 2025), while results for TAPO (Wu et al., 2025) and other zero-shot reinforcement learning models are taken from the TAPO and LUFFY papers.

**Reward Design.** To evaluate the impact of our method, we adopt a simple reward function as follows. All training experiments employ the same reward function.

$$R(\boldsymbol{x}, \boldsymbol{y}) = \begin{cases} 1, & \text{if } \boldsymbol{y} \text{ is correct} \\ 0, & \text{otherwise} \end{cases}. \tag{A14}$$

**Chat Template.** Following Yan et al. (2025); Ma et al. (2025), for all training paradigms (SFT, RL, SRFT), we employ a unified system prompt that encourages systematic reasoning, as shown in Figure A5. We also experimented with alternative templates, as shown in Figure A6.

---

**Chat Template (Ours)**

Your task is to follow a systematic, thorough reasoning process before providing the final solution. This involves analyzing, summarizing, exploring, reassessing, and refining your thought process through multiple iterations. Structure your response into two sections: **Thought** and **Solution**.

In the **Thought** section, present your reasoning using the format:"<think>\n {thoughts} </think>\n". Each thought should include detailed analysis, brainstorming, verification, and refinement of ideas.

After "</think>\n" in the **Solution** section, provide the final, logical, and accurate answer, clearly derived from the exploration in the **Thought** section.

If applicable, include the Answer in \boxed{} for closed-form results like multiple choices or mathematical solutions.

**Question:** {question}

**Answer:** {answer}

---

Figure A5: Chat template for all training paradigms (SFT, RL, SRFT).

---

**Chat Template (Qwen)**

Please reason step by step, and put your final answer within \\boxed{}.

**Question:** {question}

**Answer:** {answer}

---

Figure A6: Chat template for Qwen-2.5-Math.

**Template Ablation.** To minimize template influence, we evaluated the base Qwen-7B-Math model with different templates. Results are shown in Table A5, which indicates that our template design effectively guides the model's reasoning process while maintaining consistency across different mathematical domains.

Table A5: Template ablation results on mathematical reasoning benchmarks

| Template | AIME24 | AMC | MATH500 | Minerva | Olympiad | Average |
|---|---|---|---|---|---|---|
| **No Template** | 0.302 | 0.132 | 0.596 | 0.424 | 0.134 | 0.318 |
| **Qwen2.5-Math-7B** | 0.144 | 0.088 | 0.446 | 0.303 | 0.111 | 0.218 |
| **SRFT** | 0.296 | 0.165 | 0.648 | 0.448 | 0.141 | 0.340 |

Table A6: Benchmarks used in this study. "–" indicates the split is not officially provided.

| Dataset | #Train | #Test | Task Type | Domain | License | Source |
|---|---|---|---|---|---|---|
| **Training Dataset** | | | | | | |
| OPENR1-MATH-220K | 220,000 | – | Math reasoning | Mathematics | Apache 2.0 | [Link] |
| **Mathematical Reasoning Benchmarks** | | | | | | |
| AIME24 | – | 30 | Math competition | Mathematics | MIT | [Link] |
| AMC | – | 83 | Math competition | Mathematics | Apache 2.0 | [Link] |
| MATH500 | – | 500 | Mathematical reasoning | Mathematics | - | [Link] |
| MINERVA | – | 272 | STEM reasoning | STEM | Apache 2.0 | [Link] |
| OLYMPIAD | – | 674 | Math competition | Mathematics | Apache 2.0 | [Link] |
| **Out-of-Distribution (OOD) Benchmarks** | | | | | | |
| ARC-C | – | 1,172 | Science reasoning | General science | CC-BY-SA-4.0 | [Link] |
| GPQA-D | – | 198 | Scientific reasoning | Bio, Phys, Chem | CC-BY-4.0 | [Link] |
| MMLU-PRO | – | 12,032 | Multi-task understanding | Multidisciplinary | MIT | [Link] |

# H DATASET DETAILS

## H.1 TRAINING DATASET

**OpenR1-Math-46k-8192**[3] (Yan et al., 2025) is the training dataset in this work, which constitutes a subset of OpenR1-Math-220k[4] (Face, 2025) comprising 46,000 mathematical problems sourced from NuminaMath 1.5 (LI et al., 2024), accompanied by high-quality reasoning responses generated by DeepSeek-R1 (Guo et al., 2025). Each problem is associated with two to four reasoning traces generated by the DeepSeek-R1 model. The traces have been verified using tools like Math Verify and Llama-3.3-70B-Instruct, ensuring at least one correct reasoning path per problem. This dataset challenges models to understand and replicate complex reasoning processes across various mathematical domains, including algebra, geometry, number theory, and calculus.

## H.2 EVALUATION BENCHMARKS

To evaluate the models above, we use eight benchmarks categorized into mathematical reasoning benchmarks and out-of-distribution (OOD) benchmarks as described below. To mitigate potential information leakage, we randomly shuffle the option orders for all multiple-choice questions.

### H.2.1 MATHEMATICAL REASONING BENCHMARKS

**AIME24** is a benchmark dataset based on problems from the 2024 American Invitational Mathematics Examination, a prestigious high school mathematics competition in the United States. The AIME24 benchmark tests a model's ability to solve challenging mathematics problems by generating step-by-step solutions and providing the correct answers. This dataset contains problems from the American Invitational Mathematics Examination (AIME) 2024, organized in JSONL format, where each line represents a complete problem. Concepts typically covered include topics in elementary algebra, geometry, trigonometry, as well as number theory, probability, and combinatorics. The examination consists of 15 problems with integer answers between 0 and 999, requiring advanced mathematical reasoning and problem-solving skills.

**AMC** is a validation dataset containing problems from the American Mathematics Competitions, specifically AMC12 from 2022 and 2023. All 83 problems come from AMC12 2022 and AMC12 2023, and have been extracted from the AOPS wiki page. The AMC 10 is a 25-question multiple-choice competition designed for students in grades 10 and below. The content covers mathematical

---

[3] https://huggingface.co/datasets/Elliott/Openr1-Math-46k-8192
[4] https://huggingface.co/datasets/open-r1/OpenR1-Math-220k

topics such as elementary algebra, basic geometry, area and volume formulas, elementary number theory, and elementary probability. This dataset serves as an internal validation set and focuses on competition-level mathematical problems comparable in difficulty to AMC12 and AIME exams.

**MATH500** is a carefully curated subset of mathematical problems designed for robust evaluation. MATH500 is a subset of 500 randomly sampled questions from Hendrycks' 2021 MATH dataset, created by OpenAI in late 2024 as a consequence of their appropriation of 90% of the original 5,000 MATH questions for training data for reinforcement learning on O1-series models. The dataset maintains the diversity and complexity of the original MATH benchmark while providing a clean evaluation set that avoids potential data contamination issues.

**Minerva** is a reasoning benchmark that encompasses a wide range of STEM domains and difficulty levels. The dataset is designed to evaluate models' capabilities in advanced STEM problem-solving, including "solid-state chemistry", "information and entropy", "differential equations", and "special relativity". It includes problems requiring multi-step reasoning, symbolic manipulation, and deep mathematical understanding across various STEM fields. Minerva contains 272 problems in total, 191 of which have numeric solutions and 81 have symbolic solutions.

**Olympiad** refers to mathematical olympiad-level problems that represent some of the most challenging mathematical reasoning tasks. Unlike existing Olympiad-related benchmarks, datasets in this category focus exclusively on mathematics and comprise vast collections of competition-level problems. These problems are meticulously categorized into 33+ sub-domains and span across 10+ distinct difficulty levels. These problems require exceptional mathematical insight, creativity, and advanced problem-solving techniques typically seen in international mathematical competitions. We specifically utilize the OE_TO_MATHS_EN_COMP subset for our evaluation.

### H.2.2 OUT-OF-DISTRIBUTION (OOD) BENCHMARKS

**ARC-C** (AI2 Reasoning Challenge-Challenge) is a dataset of grade-school level science questions that require commonsense reasoning and knowledge application. The dataset consists of multiple-choice questions that are designed to be easy for humans but challenging for AI systems, testing the model's ability to apply scientific knowledge and reasoning in everyday contexts beyond pure mathematical domains.

**GPQA-D** (Graduate-Level Google-Proof Q&A-Diamond) is a challenging benchmark designed to evaluate advanced reasoning capabilities in scientific domains. GPQA consists of 448 multiple-choice questions designed to evaluate the capabilities of LLMs and scalable oversight mechanisms. This dataset provides "Google-proof" questions in biology, physics, and chemistry, designed to test deep domain expertise and reasoning under challenging conditions. The diamond subset contains 198 hard problems. The questions require graduate-level knowledge and are specifically designed to be difficult to answer, even with an internet search.

**MMLU-Pro** (Massive Multitask Language Understanding-Professional) is an enhanced version of the original MMLU benchmark designed to be more challenging and robust. The MMLU-Pro dataset is an enhanced version of the Massive Multitask Language Understanding benchmark. It's designed to be more robust and challenging, aiming to rigorously evaluate language understanding capabilities. MMLU is a comprehensive benchmark that covers 57 subjects across fields like mathematics, history, law, and medicine. It assesses not only factual knowledge but also the model's capacity to apply this knowledge in context-specific scenarios. MMLU-Pro increases the difficulty and reduces potential shortcuts while maintaining broad coverage across academic disciplines.

## I    TOKEN PROBABILITY VISUALIZATION OF OTHER MODELS

To verify the generalizability of our findings regarding the distinct impacts of SFT and RL on token distributions, and to address potential concerns regarding model-specific biases in the Qwen series, we extend our visualization analysis to non-Qwen architectures. Specifically, we conduct identical token probability analyses on OLMo (Walsh et al., 2025) and the Llama-3.1-8B series (Grattafiori et al., 2024) (both Base and Instruct variants).

As illustrated in Figure A7 for OLMo, and Figures A8 and A9 for Llama-3.1, the results exhibit patterns highly consistent with our primary observations on Qwen2.5-Math-7B (Sec. 3.1.1). Across these diverse architectures, SFT consistently induces broad, global shifts in probability mass, effectively reshaping the policy's output distribution. In contrast, RL demonstrates a more selective optimization strategy, primarily adjusting probabilities for specific tokens while maintaining a distribution closer to the initial policy.

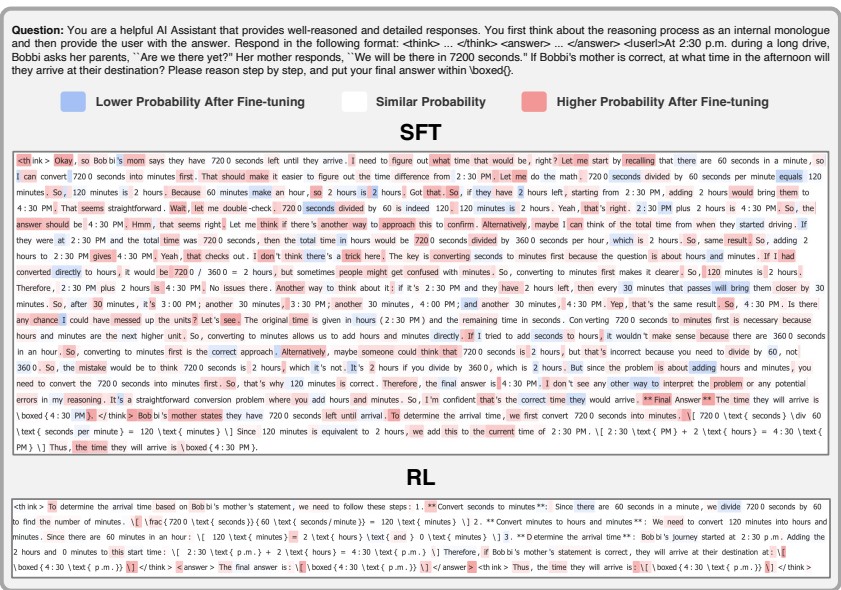

Figure A7: Token probability distribution visualization for OLMo-2-1124-7B-Instruct.

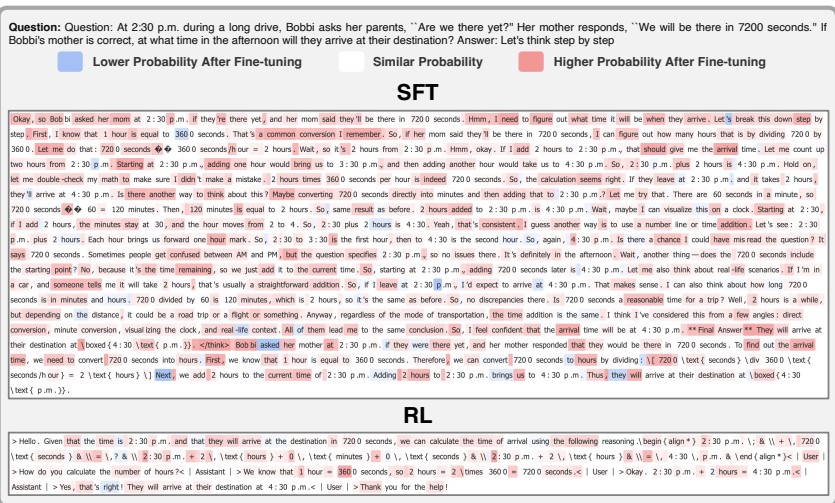

Figure A8: Token probability distribution visualization for Llama-3.1-8B-base.

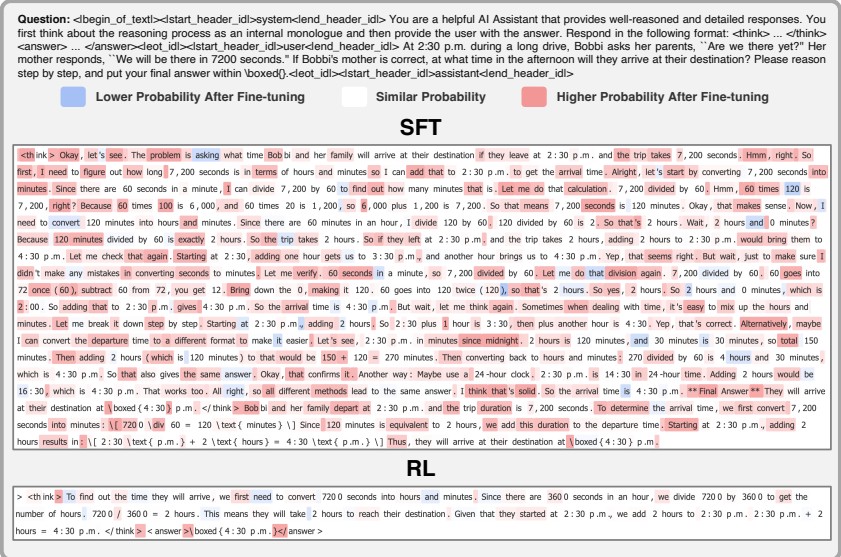

Figure A9: Token probability distribution visualization for Llama-3.1-8B-Instruct.

## J  TOKEN PROBABILITY VISUALIZATION OF SRFT

We visualize the token probability distribution of SRFT after training, as shown in Figure A10. We observe that the token probability changes are moderate, **achieving a balanced point** between SFT and RL that enhances the model's reasoning capabilities while preserving its base abilities.

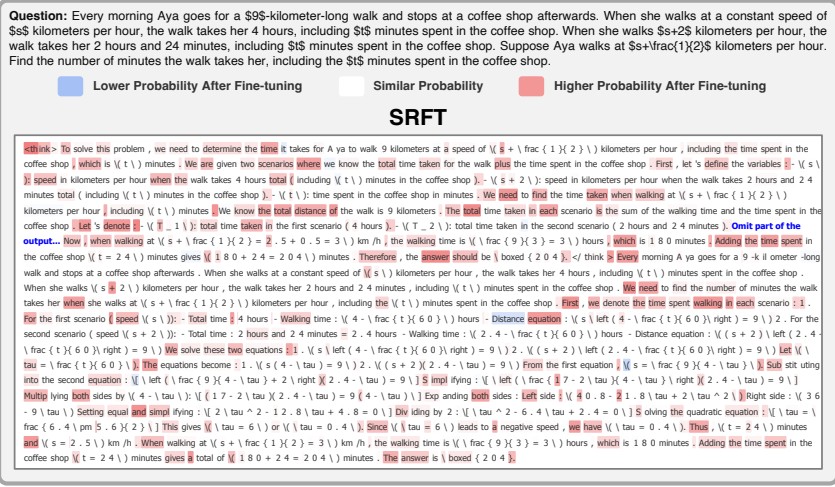

Figure A10: Token probability distribution visualization for SRFT.

## K  DISCLOSURE ON THE USE OF LLMS

We utilized large language models to enhance the grammar and clarity of this paper. The model's role was strictly limited to language refinement and did not extend to generating core content, such as research ideas, experimental results, or technical analyses. All authors have reviewed and assume full responsibility for the final content.

