# OpenReview forum: "SRFT: A Single-Stage Method with Supervised and Reinforcement Fine-Tuning for Reasoning"
_ICLR.cc/2026/Conference — ICLR 2026 Poster_

### Official Review · Reviewer_KnF6 · 2025-10-26

**Soundness:** 3
**Presentation:** 3
**Contribution:** 3
**Rating:** 6
**Confidence:** 4

**Summary:**

This paper proposes SRFT, a single-stage framework that unifies SFT and RL via an entropy-aware weighting scheme: SFT provides coarse global shifts while RL delivers fine selective refinements, coordinated by policy entropy; SRFT applies both SFT and RL on demonstrations, decomposes the RL loss on rollouts into positive/negative parts; on Qwen2.5-Math-7B across iid and ood benchmarks, SRFT improves over zero-RL baselines by 9.0% and 10.9% and maintains more stable policy entropy during training.

**Strengths:**

1. Grounded in an entropy-based view of SFT–RL integration, the paper frames SFT as a coarse, global shift and RL as a fine, selective refinement, then unifies them in a single-stage training scheme that balances both with entropy-aware weighting.
2. Methodologically clean and well-motivated: on demonstrations, SRFT applies SFT and RL jointly with an entropy-decayed SFT weight; on rollouts, it decomposes the RL loss into positive/negative components; the three parts are combined into a single, explicit loss.
3. The evaluation is comprehensive: based on Qwen2.5-Math-7B, SRFT is tested on five competition-level math benchmarks and three OOD sets, using standard metrics (avg@32 / pass@1). Table 2 shows that SRFT achieves the best in-distribution and OOD averages.
4. Training dynamics further support the design: SRFT maintains more stable policy entropy and smoother reward/length curves than RL-only training, consistent with the paper’s motivation.

**Weaknesses:**

1. Under-justified entropy weights. The entropy-aware coefficients are hard-coded to $w_{\text{SFT}}=0.5$ and $w_{\text{RL}}=0.1$. The paper notes they “can be fine-tuned” but offers no rationale or analysis for these choices.
2. Unclear compute parity in training-dynamics comparisons. SRFT optimizes three losses—demo-SFT, demo-RL, and self-rollout RL—over two data streams, trained for 500 steps with 8 rollouts per prompt. Figure 6 compares reward, response length, and entropy to RL-only, but it is not stated whether tokens, FLOPs/GPU-hours, or memory were matched, making it hard to attribute gains to the algorithm rather than extra compute.

**Questions:**

1. Justification and robustness of entropy coefficients. What motivated $w_{\text{SFT}}=0.5$ and $w_{\text{RL}}=0.1$ beyond heuristic tuning? Please include sensitivity analyses spanning orders of magnitude, or consider replacing fixed constants with a learned/monotone mapping from entropy to weights.
2. Balancing the three objectives. Since $L_{\text{SRFT}} = L_{\text{SFT}}^{\text{demo}} + L_{\text{RL}}^{\text{demo}} + L_{\text{RL}}^{\text{self-rollout}}$, have you tried introducing explicit coefficients (e.g., $\alpha$,$\beta$) and ablating their ratios to clarify relative contributions?
3. Interpreting the response-length gap in Figure 6. RL-only produces notably shorter outputs. Is this primarily because RL-only lacks a demonstration warm start while SRFT leverages expert traces (e.g., DeepSeek-R1)? If RL-only were initialized on the same demonstrations, would the reward and length curves approach SRFT?
4. Compute accounting and scalability. Please report compute and memory for SRFT vs. RL-only under the stated setup (500 steps, 8 rollouts/prompt): total and per-step FLOPs, peak memory, effective batch sizes, sequence lengths, and tokens processed. This would clarify fairness and inform scalability.

---

> ### Author Response · Authors · 2025-11-21
> **Authors Response (Part 1/2)**
>
> `Responses`
> ---
> ---
> >**Q1: Justification and robustness of entropy coefficients.**
>
> **A1:** We apologize for the lack of clarity in our initial description. We clarify that the values 0.5 and 0.1 are not fixed weights but serve as **baseline scaling coefficients** for constructing our dynamic, entropy-aware weighting mechanism (Line 875). The core of our method utilizes policy entropy $\mathcal{H}(\pi\_{\theta})$ to dynamically modulate the influence of SFT and RL losses: $w\_{\text{SFT}} = 0.5 \times \text{stop\\\_grad}(\exp(-\mathcal{H}(\pi\_{\theta})))$ and $w\_{\text{RL}} = 0.1 \times \text{stop\\\_grad}(\exp(\mathcal{H}(\pi\_{\theta})))$. Consequently, these weights evolve continuously throughout training in response to the model's uncertainty. We visualize these dynamics in **Figure A1** (Appendix A.2), showing their temporal evolution. Furthermore, we ablated these dynamic weights against fixed weights (Table 3), demonstrating that the dynamic mechanism significantly outperforms static weighting schemes. We clarify the definition and dynamic nature of these coefficients in **Section 5.2** and **Appendix A.2** of the revised manuscript.
>
> ---
> >**Q2: Balancing the three objectives.**
>
> **A2:** Thank you for highlighting the importance of balancing these objectives. While your suggestion to introduce explicit **top-level** coefficients (e.g., $\alpha, \beta$) aligns with our goal of controlling loss contributions, our approach embeds these coefficients **dynamically** within the individual loss terms ($L_{\text{SFT}}^{\text{demo.}}$ and $L_{\text{RL}}^{\text{self-rollout}}$). This design ensures the **relative contribution** is regulated by the model's policy entropy rather than static manual tuning. To further clarify the specific contribution of each component, we conducted a **leave-one-out ablation study**. The results are presented below:
>
> | Method | AIME24 | AMC | MATH-500 | Minerva | Olympiad | Avg. |
> | :--- | :--- | :--- | :--- | :--- | :--- | :--- |
> | w/o $\text{demo}_\text{SFT}$ (Remove Loss) | 29.5 | 65.2 | 85.6 | 35.9 | 53.1 | 53.9 |
> | w/o $\text{demo}_\text{RL}$ (Remove Loss) | 30.8 | 68.0 | 86.4 | 36.2 | 54.5 | 55.2 |
> | SRFT | 35.3 | 74.3 | 89.8 | 39.7 | 58.3 | 59.5 |
>
> In this experiment, w/o $\text{demo}\_\text{SFT}$ removes the SFT loss on demonstrations, and w/o $\text{demo}\_\text{RL}$ removes the RL loss on demonstrations. It is worth noting that, as our method is fundamentally built upon the RL framework (Line 273), the self-rollout RL component ($\mathcal{L}_\text{RL}^\text{self-rollout}$) is consistently retained. The experimental results demonstrate that these multiple components contribute synergistically to the performance of SRFT. We have incorporated this analysis into **Section 5.2** of the revised paper.
>
> ---
> >**Q3: Interpreting the response-length gap in Figure 6.**
>
> **A3:** Yes, initializing RL with SFT indeed **bridges the response length gap**. To investigate this, we add the response length and reward curves for the **SFT$\rightarrow$RL** baseline in the revised manuscript. When RL is initialized from an SFT model, the response length starts and maintains a high level ($\approx$ 4k tokens), exceeding SRFT's length ($\approx$ 3k tokens). Despite the longer responses, the performance of SFT$\rightarrow$RL exhibits premature convergence. In contrast, SRFT sustains continuous performance improvement. This indicates that SRFT's advantage stems from its **single-stage integration of demonstration and rollout data**, which fosters more effective exploration than simply initializing with long-context SFT models. These results and visualizations have been included in **Appendix A.4** of the revised paper.

---

> ### Author Response · Authors · 2025-11-21
> **Authors Response (Part 2/2)**
>
> >**Q4: Compute accounting and scalability.**
>
> **A4:** Thank you for the suggestion. We provide a detailed breakdown of the computational costs for pure RL, SFT$\rightarrow$RL, and SRFT in the table below:
>
> | Compute | Pure RL | SFT $\to$ RL| SRFT |
> | :--- | :--- | :--- | :--- |
> | **Effective Batch Size** | 1024 | 1024 | 1024 |
> | **Tokens Processed / Step** | $\approx 1.1 \times 10^7$ | $\approx 2.5 \times 10^7$ | $\approx 2.0 \times 10^7$ |
> | **FLOPs per Step** | $7.7 \times 10^{16}$ | $1.8 \times 10^{17}$ | **$1.4 \times 10^{17}$** |
> | **Total FLOPs** | $3.9 \times 10^{19}$ | $9.9 \times 10^{19}$ | **$7.1 \times 10^{19}$** |
> | **Peak Memory** | $\approx$ 42GB | $\approx$ 78GB | $\approx$ 74GB |
> | **Accuracy** | 49.3% | 54.5% | 59.5% |
>
> **Analysis:**
> * **SRFT vs. Pure RL:** The computational cost of SRFT is indeed higher than pure RL. However, we respectfully note that this increase is primarily driven by the model generating significantly **longer response lengths** (approx. 3k tokens for SRFT vs. 1k for RL) rather than algorithmic overhead. The model learns to reason more thoroughly, which naturally consumes more compute during rollouts.
> * **SRFT vs. SFT$\rightarrow$RL:** Comparison with the SFT$\rightarrow$RL pipeline shows that SRFT achieves a better trade-off between performance and computational cost (lower FLOPs/Memory than SFT$\rightarrow$RL, yet higher accuracy).
>
> Besides, we also note that comparing compute costs across methods is non-trivial, as the ratio of training vs. sampling vs. log-prob computation costs varies significantly depending on the **specific implementation and infrastructure**.
>
>
> `Summary`
> ---
> ---
> Thank you for your valuable review and constructive feedback. We appreciate your recognition that our work is **methodologically clean and well-motivated**, that **the evaluation is comprehensive**, and that **the results effectively support our design and motivation**.
>
>
> We are also encouraged by the shared affirmations from Reviewers R2 (svi3), R3 (5rC4), and R4 (KnF6) regarding:
>
> - **Motivation and Novelty**: (**R1**: *The proposal ... is interesting*), (**R2**: *The motivation to introduce the SRFT method is well-grounded*), (**R3**: *Motivation is clear*).
> - **Methodological Design**: (**R1**: *Unify SFT and RL into a single stage, enables finer control over the trade-off between the two*).
> - **Empirical Evidence and Analysis**: (**R1**: *The paper offers a comprehensive analysis of learning dynamics...*), (**R2**: *Based on the empirical analyses*), (**R3**: *Evaluation on both in-distribution and out-of-distribution benchmarks is good... multiple ablations are presented*).
> - **Clarity and Presentation**: (**R1**: *The manuscript is clearly written and easy to follow*), (**R2**: *The paper is well written and easy to follow*), (**R3**: *Evaluation methodology is thoroughly reported, figures and Tables are well made*).
>
> We hope that our responses have fully addressed your concerns, and we welcome any further suggestions and discussions. **If our responses have resolved your concerns, we would sincerely appreciate your consideration for raising the score.**

---

> ### Author Response · Authors · 2025-11-27
>
> Dear Reviewer,
>
> We are writing to follow up on our response to your valuable feedback. We hope that our clarifications, along with the revisions in the updated manuscript, have addressed your concerns. Assuming this is the case, we kindly ask that you consider re-evaluating your score. Otherwise, please let us know if you have any further questions.
>
> Thank you for your time and effort in reviewing our paper.
>
> Best regards,
>
> All Authors of Submission #1626

---

### Official Review · Reviewer_5rC4 · 2025-10-30

**Soundness:** 3
**Presentation:** 4
**Contribution:** 3
**Rating:** 6
**Confidence:** 3

**Summary:**

The paper introduces SRFT, a single-stage fine-tuning paradigm that combines supervised fine-tuning (SFT) and Reinforcement Learning (RL) through an entropy-aware weighting mechanism. Instead of applying SFT and RL sequentially, SRFT jointly optimizes both objectives. The approach introduces a combined Loss function that integrates SFT, off-policy RL and on-policy RL components. The authors analyse the resulting learning dynamic and compare singe-stage versus two-stage training. The experiments are based on Qwen2.5-Math-7B, with ablations on two additional Qwen variants and show that SRFT outperforms standard SFT->RL pipelines.

**Strengths:**

- Evaluation methodology is thoroughly reported and appears to follow best practices (e.g. multi-seed evaluation on high-variance benchmarks, ablation across prompt variants) which increases confidence in the robustness of the results.
- Evaluation on both in-distribution and out-of-distribution benchmarks is good.
- The authors include a thorough analysis of learning dynamics.
- Multiple ablations are presented, including on entropy weighting factors and training on different Qwen-based model variants
- Figures and Tables are well made
- Motivation is clear

**Weaknesses:**

- The method is only tested on the Qwen model family.
- The claims in section 3.1 seem to lack quantitative validation. (It is the result of one training run per category on Qwen2.5-Math-7B)

**Questions:**

- **Q1:** Training stability
   - (a) Reinforcement learning on smaller models (1.5B/7B) is often brittle and sensitive to hyperparameters. Could the authors comment on the training stability of SRFT compared to standard RL (e.g., GRPO)?
   - (b) Introducing additional loss components can sometimes destabilize optimization. How does SRFT’s stability compare to pure SFT training in practice?
- **Q2:** It is mentioned that math-verify is used for dataset filtering. Is it also used for the answer matching in the evaluation? If not, could the authors clarify which evaluation framework or procedure is used?
- **Q3:** Generalization across models
   - (a) Does SRFT also improve performance when starting from an RL tuned model (like Qwen2.5-Math-7B-Instruct)?
   - (b) Were other model architectures tested (e.g. Gemma, LLaMa) to asses the whether SRFT benefits generalize beyond the Qwen family?
- **Q4:** How does the computational cost of SRFT compare to the standard SFT->RL pipeline?

---

> ### Author Response · Authors · 2025-11-21
> **Authors Response (Part 1/2)**
>
> `Responses`
> ---
> ---
> >**Q1: The claims in Section 3.1 seem to lack quantitative validation.**
>
> **A1:** We supplement our analysis with **additional validation** in two key aspects:
>
> * **Token Distribution:** We add visualization experiments for the **Llama-3.1** and **OLMo** model series, as shown in **Appendix I** of the revised manuscript. The results align closely with the phenomena observed on Qwen in the original paper. Furthermore, Figure 2(b) presents statistical results aggregate across all questions in five benchmarks (comprising approximately **600,000** token probability samples), which we respectfully submit demonstrates statistical significance.
> * **Learning Dynamics:** We include experiments with varying SFT epochs, as detailed in **Appendix C.3** of the revised manuscript. The results indicate that SFT induces a larger distributional shift compared to RL, and while varying epochs can modulate overfitting, determining the optimal SFT duration remains an underexplored challenge [1]. In contrast, our SRFT method utilizes a single-stage combination of SFT and RL with adaptive weights to **achieve direct optimization** of the LLM policy.
>
> ---
> >**Q2: Training stability of SFT, RL, and SRFT.**
>
> **A2:**
> * **Compared to Pure SFT:** We acknowledge that the **RL-based SRFT framework** introduces complexity compared to the **stable, convex-like optimization landscape** of pure SFT (via Maximum Likelihood Estimation). However, **theoretically**, SRFT employs entropy-aware weighting to dynamically modulate loss components based on policy uncertainty, effectively managing the SFT-RL trade-off. **In practice**, SRFT demonstrates entropy stability (Figure 6) and superior performance on both in-distribution and out-of-distribution benchmarks compared to SFT, successfully striking a balance between stability and performance.
> * **Compared to Pure RL:** RL training stability is a critical yet not fully resolved research area [2][3][4]. **Theoretically**, SRFT incorporates two entropy-aware weighting mechanisms specifically designed to mitigate instability arising from demonstration data and positive RL rollout samples (Lines 310 and 346). **Technically**, given the vast hyperparameter space (e.g., the `verl` framework contains dozens of RL-related parameters) and the inherent implementation sensitivities [5][6], like similar RL for reasoning works [7][8], we do not perform an exhaustive hyperparameter stability search. Besides, **empirically**, as shown in the training dynamics (Figure 6), we observe that neither SRFT nor Pure RL exhibited training collapse within the 500-step horizon.
>
> ---
> >**Q3: Is Math-verify also used for the answer matching in the evaluation?**
>
> **A3:** Yes. We utilize Math-Verify for answer matching (Line 1282 of the original manuscript).
>
> ---
> >**Q4: Does SRFT also improve performance when starting from an RL tuned model (like Qwen2.5-Math-7B-Instruct)?**
>
> **A4:** Yes. We include evaluation results starting from **Qwen2.5-7B-Instruct** in the original manuscript (**Table A1**). The results shows that SRFT continues to demonstrate performance improvements when initialized from a fine-tuned model.
>
> ---
> >**Q5: Were other model architectures tested to assess whether SRFT benefits generalize beyond the Qwen family?**
>
> **A5:** We add experiments using the **Llama-3.1-8B** model. The results are presented below:
>
> | Model | AIME24 | AMC | MATH-500 | Minerva | Olympiad | Avg. |
> | :--- | :--- | :--- | :--- | :--- | :--- | :--- |
> | Llama-3.1-8B | 0.3 | 4.2 | 13.8 | 4.4 | 3.8 | 5.3 |
> | SFT | 0.5 | 5.4 | 20.2 | 4.0 | 5.3 | 7.1 |
> | GRPO | 0.3 | 9.4 | 23.4 | 17.6 | 6.1 | 11.4 |
> | LUFFY | 1.9 | 13.5 | 39.0 | 15.1 | 9.6 | 15.8 |
> | SRFT | 1.9 | 14.3 | 40.1 | 15.3 | 9.5 | 16.2 |
>
> These results demonstrate that SRFT improves performance on non-Qwen architectures as well. We incorporate these findings into **Appendix A.1** of the revised manuscript.
>
> ---
> >**Q6: How does the computational cost of SRFT compare to the standard SFT->RL pipeline?**
>
> **A6:** Thank you for this question. SRFT reduces overall training costs by **amortizing** the explicit SFT stage of the sequential pipeline into the single-stage RL process. Quantitatively, under our experimental settings (Batch Size=128, Dataset $\approx$ 46k, SFT baseline=3 epochs), the computational cost of SRFT at 500 steps is approximately **0.8x** that of the SFT$\rightarrow$RL pipeline. Notably, at this 500-step mark, SRFT already yields a higher average performance (+0.5 points) than the completed SFT$\rightarrow$RL baseline. The total compute cost of SRFT only reaches parity with the sequential baseline at approximately 1100 steps, highlighting the significant computational efficiency of our method.

---

> ### Author Response · Authors · 2025-11-21
> **Authors Response (Part 2/2)**
>
> `Summary`
> ---
> ---
> Thank you for your valuable review and constructive feedback. We appreciate your recognition of **the thoroughly reported evaluation methodology**, **the good results on both ID and OOD benchmarks**, **comprehensive analysis and multiple ablations**, **the high quality of the figures and tables**, and **the clear motivation** of our work.
>
> We are also encouraged by the shared affirmations from Reviewers R2 (svi3), R3 (5rC4), and R4 (KnF6) regarding:
>
> - **Motivation and Novelty**: (**R1**: *The proposal ... is interesting*), (**R2**: *The motivation to introduce the SRFT method is well-grounded*), (**R4**: *Methodologically clean and well-motivated*).
> - **Methodological Design**: (**R1**: *Unify SFT and RL into a single stage, enables finer control over the trade-off between the two*), (**R4**: *Grounded in an entropy-based view of SFT–RL integration*).
> - **Empirical Evidence and Analysis**: (**R1**: *The paper offers a comprehensive analysis of learning dynamics...*), (**R2**: *Based on the empirical analyses*), (**R4**: *The evaluation is comprehensive, training dynamics further support the design*).
> - **Clarity and Presentation**: (**R1**: *The manuscript is clearly written and easy to follow*), (**R2**: *The paper is well written and easy to follow*).
>
> We hope that our responses have fully addressed your concerns, and we welcome any further suggestions and discussions. **If our responses have resolved your concerns, we would sincerely appreciate your consideration for raising the score.**
>
> `References`
> ---
> ---
> [1] "RL fine-tuning heals ood forgetting in SFT." arXiv:2509.12235 (2025).
>
> [2] "Part I: Tricks or traps? a deep dive into RL for LLM reasoning." arXiv:2508.08221 (2025).
>
> [3] "ProRL: Prolonged reinforcement learning expands reasoning boundaries in large language models." arXiv:2505.24864 (2025).
>
> [4] "The art of scaling reinforcement learning compute for LLMs." arXiv:2510.13786 (2025).
>
> [5] "Your efficient RL framework secretly brings you off-policy RL training" Blog (2025).
>
> [6] "Defeating the Training-Inference Mismatch via FP16."arXiv:2510.26788 (2025).
>
> [7] "DAPO: An open-source LLM reinforcement learning system at scale." NeurIPS (2025).
>
> [8] "Understanding r1-zero-like training: A critical perspective." COLM (2025).
>
> [9] "Group sequence policy optimization." arXiv:2507.18071 (2025).

---

> ### Author Response · Authors · 2025-11-27
>
> Dear Reviewer,
>
> We are writing to follow up on our response to your valuable feedback. We hope that our clarifications, along with the revisions in the updated manuscript, have addressed your concerns. Assuming this is the case, we kindly ask that you consider re-evaluating your score. Otherwise, please let us know if you have any further questions.
>
> Thank you for your time and effort in reviewing our paper.
>
> Best regards,
>
> All Authors of Submission #1626

---

> ### Comment · Reviewer_5rC4 · 2025-11-27
> **Response to Authors' Rebuttal**
>
> I thank the authors for conducting additional experiments and for addressing my questions. I seemed to have missed a detail in Table 1, the Qwen2.5-7B-Math results appear unusually low compared to those reported in [1] (Table 3). This makes the gains across the different training variants appear marginal. A clarification from the authors would be helpful.
>
> Given the new LLaMA results, I am not fully convinced that the proposed method offers a strong advantage over existing approaches. That said, math fine-tuning may be a suboptimal regime for LLaMA due to its pre-training data being less math-focused than Qwen's. I would appreciate any intuition from the authors on this point. The Qwen results are quite strong, however, which makes me lean towards acceptance. Consequently, and given the remaining concerns, I will maintain my original score.
>
> [1]https://arxiv.org/abs/2504.07086

---

> > ### Author Response · Authors · 2025-11-27
> > **Thank you for your follow-up**
> >
> > We sincerely thank you for engaging with us. Below, we address your remaining concerns regarding the baseline discrepancies and the Llama results. We hope these clarifications can ease your concerns.
> >
> > - **Discrepancy in Qwen2.5-7B-Math Baselines**: The discrepancy stems from **different** evaluation protocols. Our original evaluation followed the settings used in prior works like [1][2][3][4], which yielded results consistent with our paper. To address your concern and eliminate ambiguity, we re-evaluate SRFT and key baselines using the same evaluation protocols suggested by Sober Reasoning. As shown in the table below, under this protocol, SRFT still maintains a **substantial advantage** (avg. +18.0 points over Base and +7.4 points over RL$_{\text{GRPO}}$). This confirms that the gains reported in our paper are robust and not artifacts of a specific evaluation setting.
> >
> >     | Model | AIME24 | AMC | MATH-500 | Minerva | Olympiad | Avg. |
> >     | :--- | :--- | :--- | :--- | :--- | :--- | :--- |
> >     | Qwen2.5-Math-7B | 20.7 $\pm$ 3.8 | 56.2 $\pm$ 5.7 | 64.3 $\pm$ 0.5 | 17.3 $\pm$ 1.9 | 29.0 $\pm$ 0.5 | 37.5 |
> >     | RL$_\text{GRPO}$ | 22.5 $\pm$ 2.6 | 64.4 $\pm$ 3.4 | 78.6 $\pm$ 0.3 | 32.8 $\pm$ 1.3 | 42.3 $\pm$ 0.9 | 48.1 |
> >     | SRFT | 28.3 $\pm$ 2.4 | 71.8 $\pm$ 3.9 | 88.0 $\pm$ 0.2 | 34.4 $\pm$ 1.2 | 54.7 $\pm$ 0.5 | 55.5 |
> >     | Performance Gain | +7.6 | +15.6 | +23.7 | +17.1 | +25.7 | +18.0 |
> >
> > - **Intuition on Llama Results**: Yes, we agree with your intuition that Llama is generally a **suboptimal starting point** for math tasks compared to Qwen [5][6] due to differences in pre-training data distributions. The core motivation of SRFT is to unify SFT and RL into a single-stage framework, building upon the analysis in Section 3. As demonstrated in our additional analysis in Appendix I, the distributional dynamics of SFT and RL on Qwen are **consistent with** those on Llama and OLMo. Therefore, while Llama's absolute performance ceiling is lower, we respectfully argue that the methodological advantage of SRFT can generalize across different model families.
> >
> > `References`
> > ---
> > ---
> >
> > [1] "Learning to reason under off-policy guidance." NeurIPS (2025).
> >
> > [2] "Towards a unified view of large language model post-training." arXiv:2509.04419 (2025).
> >
> > [3] "Thought-augmented policy optimization: Bridging external guidance and internal capabilities." arXiv:2505.15692 (2025).
> >
> > [4] "Blending supervised and reinforcement fine-tuning with prefix sampling." arXiv:2507.01679 (2025).
> >
> > [5] "Beyond the 80/20 rule: High-entropy minority tokens drive effective reinforcement learning for LLM reasoning." NeurIPS (2025).
> >
> > [6] "Octothinker: Mid-training incentivizes reinforcement learning scaling." arXiv:2506.20512 (2025).

---

### Official Review · Reviewer_svi3 · 2025-11-01

**Soundness:** 2
**Presentation:** 3
**Contribution:** 2
**Rating:** 4
**Confidence:** 3

**Summary:**

This paper conducts an analysis on the learning dynamics (as evidenced by induced token probability distributions, entropy-based analysis and training dynamics) of SFT and RL approaches for fine-tuning LLMs for improved reasoning. The paper shows that SFT induces coarse-grained and large changes in the base policy distribution of the LM whereas RL induces finer-grained and localized policy changes. Using these observations, the paper introduces a joint single-stage method called SRFT that couples three different loss objectives including self-rollouts and standard SFT + RL. Results showcase that the proposed method improves results across several benchmarks like AIME'24, AMC'23 and even OOD benchmarks like GPQA-D and MMLU-Pro.

**Strengths:**

- The paper is well written and easy to follow.
- The analyses conducted are presented in a compact form making it easily digestible.
- The motivation to introduce the SRFT method is well-grounded based on the empirical analyses conducted in the paper.

**Weaknesses:**

- Currently in line 160, there is a claim that states that the fig 2a results reveal a fundamental difference between SFT and RL regarding their reshaped probability distributions. Could there be some details provided regarding how exactly the heat-map is computed. Currently, it is unclear and a lot of details are omitted.
    - Is the token output sequence generated by the base model or the SFT/RLd model? Further, are the log-probs simply computed token-wise with the base and the SFT/RLd model? The appendix mentions teacher forcing, but it is unclear what that means in this context, is the base model the teacher whose rollouts are used?
    - This experiment is also only done with the Qwen model series, since recent works have shown that Qwen models might have some unintended biases (https://arxiv.org/abs/2506.10947, https://arxiv.org/pdf/2507.10532), it would be great to also show the result on non-Qwen models.
    - It would be interesting to further establish this trend by plotting KL-divergence / JSD plots wrt to the base and SFT/RL models.
    - Further, this heat-map is for one particular problem instance. It would be good to show some more examples, even though fig 2b is a more a quantitative treatment of the same.
- The numbers shown in tab 1 don’t seem fully accurate for Qwen-2.5-Math-7B, AMC and Minerva. Reference the Sober reasoning paper (https://arxiv.org/pdf/2504.07086). What hyper parameters were used for evaluation? Would the same takeaways hold If the "optimal" evaluation hyper parameters were used as suggested in the sober reasoning paper?
- In lines 202-203, it is stated: “This result suggests that SFT may overfit to demonstrations, and the subsequent RL attempts to correct this deviation, thereby increasing the learning tax of the two-stage paradigm.” Is this result globally true or does this depend on the number of epochs of SFT training (currently the paper seems to be doing 3 epochs). It would be interesting to ensure that this result globally holds true regardless of the number of optimization steps (since this is important for final performance as noted in OpenThoughts (https://arxiv.org/abs/2506.04178))
- Is the SRFT method better than simple on-policy distillation (as shown in methods like GKD: https://arxiv.org/abs/2306.13649). For a fair comparison in the current setting, the teacher used for on-policy distillation should be DeepSeek-R1.
- All experiments are only conducted with Qwen models. As mentioned earlier in the review, prior works have shown biases and data contamination in Qwen models with regards to the benchmarks used for testing. Would the results of SRFT also hold for non-Qwen models?
- Why do the performance numbers between tab 1 and tab 2 differ for qwen2.5-math-7B, shouldn’t they be the same numbers?
- There should be an ablation with different weighting factors for each of the loss terms in eq 12. For example, it might be plausible that only the self-rollout RL loss term might be the most beneficial (similar to an on-policy distillation loss), hence a leave-one-out ablation experiment (where each time the training is done only with two loss terms in the mix is used) should be conducted for more conclusive results.
- Do the results also improve on AIME’25? This is an important test of generalisation as shown in Sober reasoning paper.

**Questions:**

All my questions are also raised in the weakness section so that its easy for the authors to answer all concerns / queries jointly.

---

> ### Author Response · Authors · 2025-11-21
> **Authors Response (Part 1/3)**
>
> `Responses`
> ---
> ---
> >**Q1: More details and results provided for Figure 2.**
>
> **A1.1: Heatmap Visualization Details:** We add the following details to **Appendix B.1** in the revised manuscript:
> * **Source of Output Sequences:** Both response sequences shown in Figure 2(a) are generated independently by the **fine-tuned SFT and RL models**, respectively, given the same prompt input. We do not use the base model's generation because, for many complex tasks, the base model produces **incorrect or overly brief responses**, which are unsuitable for visualizing subtle probability shifts. Using the fine-tuned model's own generation path allows us to observe how fine-tuning reallocates probability mass along its preferred trajectory relative to the base model.
> * **Token-wise Probability Calculation:** For every token $y_t$ in the generated sequence, given the prefix $y_{<t}$, we compute two conditional probabilities: the fine-tuned probability $p_\text{SFT/RL}(y_t | y_{<t})$ and the base model probability $p_\text{base}(y_t | y_{<t})$. Note that while the sequence is generated by the fine-tuned model, the probabilities are queried from both models using identical contexts.
> * **Color Coding:** The color of each token is determined by the probability shift $\Delta_p = p_\text{SFT/RL}(y_t | y_{<t}) - p_\text{base}(y_t | y_{<t})$. Red ($\Delta_p > 0$) indicates the fine-tuning amplified the token's probability; Blue ($\Delta_p < 0$) indicates suppression; and Grey/Colorless ($\Delta_p \approx 0$) indicates negligible change. This visualization allows us to observe macro-level probability migration trends while identifying specific token-level shifts.
>
> **A1.2: Additional Architecture Visualizations:** To further enhance the robustness of our visualization findings, we add heatmaps for three non-Qwen architectures: **Llama-3.1-Base**, **Llama-3.1-Instruct**, and **OLMo-2-Instruct**. As shown in the revised manuscript **Appendix I**, these models exhibit similar patterns to Qwen, reinforcing the generalizability of our observations regarding SFT's global shifts versus RL's selective adjustments.
>
> **A1.3: KL-Divergence Analysis:** In the original manuscript (Figure 3, Line 190), we visualize the KL-divergence trends between the **training models and three reference models** (including the base model Qwen2.5-Math-7B). The results quantitatively show that SFT induces a significantly larger distributional shift from the base model compared to RL, aligning with our heatmap observations.
>
> **A1.4: Clarification on Figure 2(b):** We apologize for any confusion. Figure 2(b) represents a statistical aggregation of results across **five benchmarks** (comprising approximately **600,000** token probability samples), rather than a quantitative treatment of a single problem instance (Line 128). We clarify this in the revised manuscript for Figure 2(b).
>
> ---
> >**Q2: Evaluation settings in Table 1.**
>
> **A2:**
> * **Evaluation Methodology:** We appreciate the reference to *Sober Reasoning* [1]. We agree that hyperparameters can significantly impact performance. While we do not perform an exhaustive hyperparameter grid search for every baseline due to the prohibitive computational cost of re-evaluating all methods, we ensure a **fair comparative analysis** by running all baselines under identical evaluation settings: unified prompt templates, the same reward function, and consistent inference parameters (Lines 370 and 1282). This **controlled variable approach** ensures that performance differences are attributable to the training algorithm itself [2][3][4]. The above analysis is added to the revised **Appendix G**.
> * **Robustness Check:** To verify our conclusions, we re-evaluate SRFT and key baselines using the **evaluation protocols** suggested by *Sober Reasoning*. As shown below, SRFT maintains **a substantial advantage** (avg. +7.4 points over RL_GRPO). Furthermore, as shown in Figure 6, SRFT exhibits superior training stability and faster convergence, intrinsic advantages that persist regardless of inference parameters.
>
>     | Model | AIME24 | AMC | MATH-500 | Minerva | Olympiad | Avg. |
>     | :--- | :--- | :--- | :--- | :--- | :--- | :--- |
>     | Qwen2.5-Math-7B | 20.7 $\pm$ 3.8 | 56.2 $\pm$ 5.7 | 64.3 $\pm$ 0.5 | 17.3 $\pm$ 1.9 | 29.0 $\pm$ 0.5 | 37.5 |
>     | RL$_\text{GRPO}$ | 22.5 $\pm$ 2.6 | 64.4 $\pm$ 3.4 | 78.6 $\pm$ 0.3 | 32.8 $\pm$ 1.3 | 42.3 $\pm$ 0.9 | 48.1 |
>     | SRFT | 28.3 $\pm$ 2.4 | 71.8 $\pm$ 3.9 | 88.0 $\pm$ 0.2 | 34.4 $\pm$ 1.2 | 54.7 $\pm$ 0.5 | 55.5 |
>
> * **Correction for Table 1:** Thank you for catching this. The discrepancy in Table 1 was indeed a typo. We correct Table 1 in the revision to be consistent with Table 2. This correction does not alter the conclusion regarding SRFT's relative performance gain over baselines.

---

> ### Author Response · Authors · 2025-11-21
> **Authors Response (Part 2/3)**
>
> >**Q3: Impact of different SFT epochs on Section 3.2 statement.**
>
> **A3:** Our original manuscript state that "SFT *may* overfit to demonstrations, and the subsequent RL attempts to correct this deviation, thereby increasing the learning tax of the two-stage paradigm.",regarding the case of fewer SFT epochs, we conduct an analysis from two angles: **existing empirical results** and **supplementary experiments**.
> * **Existing Evidence:** As shown in Figure 3, early-stage SFT moves the policy toward a local optimum, but extended SFT causes it to deviate significantly. We hypothesize that early stopping might mitigate this, but identifying the optimal stopping point is **non-trivial**. Recent literature [5][6] supports the view that overfitting in SFT can hinder subsequent RL optimization.
> * **Supplementary Evidence:** To investigate this, we add an analysis of different SFT epochs in **Appendix C.3** (Figure A3). The results confirm that while early stopping (e.g., 1 epoch) reduces the deviation compared to extended SFT, the "learning tax" persists. SRFT addresses this dilemma by unifying SFT and RL into a single stage with adaptive weighting, thereby **optimizing the policy directly** without the need for manual stage-switching or complex hyperparameter tuning for SFT epochs.
>
> ---
> >**Q4: Is SRFT better than simple on-policy distillation (e.g., GKD)?**
>
> **A4:** We respectfully clarify that a direct "apple-to-apple" comparison between SRFT and on-policy distillation (like GKD) is challenging due to fundamental theoretical and technical differences:
>
> | Feature | SRFT (Ours) | GKD (On-Policy Distillation) |
> | :--- | :--- | :--- |
> | **Framework** | **Reinforcement Learning:** Optimizes policy directly via reward maximization. Can explore beyond the teacher. | **Knowledge Distillation:** Minimizes distribution mismatch between student and teacher. Theoretically bounded by teacher performance. |
> | **Optimization** | Can discover solutions superior to the demonstrations via exploration. | Aims to mimic the teacher; typically does not surpass the teacher's capabilities. |
> | **Vocabulary** | **No constraint:** Can use offline data from any model (e.g., DeepSeek-R1) regardless of tokenizer. | **Strict constraint:** Teacher and student must share the same tokenizer/vocabulary for Logit KD. |
> | **Requirement** | Can start directly from a **Base Model**. | Typically requires an **SFT-initialized model** capable of generating adequate sequences for feedback. |
> | **Teacher Access** | Offline demonstrations (efficient). | Requires online access to the teacher model during training to compute forward logits (computationally expensive). |
>
> Given the vocabulary mismatch between our teacher (DeepSeek-R1) and student (Qwen-2.5-Math), applying standard GKD would require additional vocabulary alignment techniques and local deployment of the teacher model. We are actively attempting to implement this on-policy distillation experiment to provide a direct comparison if feasible. Furthermore, we add this detailed discussion to **Appendix D**.
>
> ---
> >**Q5: Do SRFT results hold for non-Qwen models?**
>
> **A5:** Yes. We add experiments using **Llama-3.1-8B** in **Appendix A.1**. As shown below, SRFT achieves consistent improvements over SFT and RL baselines on the non-Qwen model.
>
> | Model | AIME24 | AMC | MATH-500 | Minerva | Olympiad | Avg. |
> | :--- | :--- | :--- | :--- | :--- | :--- | :--- |
> | Llama-3.1-8B | 0.3 | 4.2 | 13.8 | 4.4 | 3.8 | 5.3 |
> | SFT | 0.5 | 5.4 | 20.2 | 4.0 | 5.3 | 7.1 |
> | GRPO | 0.3 | 9.4 | 23.4 | 17.6 | 6.1 | 11.4 |
> | LUFFY | 1.9 | 13.5 | 39.0 | 15.1 | 9.6 | 15.8 |
> | SRFT | 1.9 | 14.3 | 40.1 | 15.3 | 9.5 | 16.2 |
>
> ---
> >**Q6: Do results improve on AIME’25?**
>
> **A6:** Yes. We add the experimental evaluation of AIME25 in the revised version of **Appendix A.1**, as shown below:
>
> |  | Qwen2.5-Math-7B | SFT | GRPO | SFT->RL | SFT+RL | SRFT |
> | --- | --- | --- | --- | --- | --- | --- |
> | **AIME25** | 4.9 | 20.3 | 14.5 | 21.2 | 15.6 | 21.6 |
>
> The results indicate that SRFT still maintains performance improvement on this benchmark.

---

> ### Author Response · Authors · 2025-11-21
> **Authors Response (Part 3/3)**
>
> >**Q7: Leave-one-out ablation experiment.**
>
> **A7:** Thank you for your suggestion. We conduct leave-one-out ablation experiments with key modules removed, and the results are shown below:
>
> |  | AIME24 | AMC | MATH-500 | Minerva | Olympiad | Avg. |
> | --- | --- | --- | --- | --- | --- | --- |
> | w/o $\text{demo}_\text{SFT}$ (Remove Loss) | 29.5 | 65.2 | 85.6 | 35.9 | 53.1 | 53.9 |
> | w/o $\text{demo}_\text{RL}$ (Remove Loss) | 30.8 | 68.0 | 86.4 | 36.2 | 54.5 | 55.2 |
> | SRFT | 35.3 | 74.3 | 89.8 | 39.7 | 58.3 | 59.5 |
>
> Among them, w/o $\text{demo}\_\text{SFT}$ removes the SFT on demonstration data, and w/o $\text{demo}\_\text{RL}$ removes the RL on demonstration data. It is worth noting that, as our method is fundamentally built upon the RL framework (Line 279), we exclude the ablation of $\mathcal{L}\_\text{RL}^\text{self-rollout}$, meaning that the online RL training component was consistently retained. The experimental results demonstrate that these multiple components contribute jointly to the performance, validating the effectiveness of SRFT. These results and the corresponding analysis are incorporated into **Section 5.2** of the revised manuscript.
>
>
> `Summary`
> ---
> ---
> Thank you for your valuable review and constructive feedback. We appreciate your recognition of **the digestible analysis**, **the well-grounded motivation**, and **the clarity of writing**.
>
> We are also encouraged by the shared affirmations from Reviewers R2 (svi3), R3 (5rC4), and R4 (KnF6) regarding:
>
> - **Motivation and Novelty**: (**R1**: *The proposal ... is interesting*), (**R3**: *Motivation is clear*), (**R4**: *Methodologically clean and well-motivated*).
> - **Methodological Design**: (**R1**: *Unify SFT and RL into a single stage, enables finer control over the trade-off between the two*), (**R4**: *Grounded in an entropy-based view of SFT–RL integration*).
> - **Empirical Evidence and Analysis**: (**R1**: *The paper offers a comprehensive analysis of learning dynamics...*), (**R3**: *Evaluation on both in-distribution and out-of-distribution benchmarks is good... multiple ablations are presented*), (**R4**: *The evaluation is comprehensive, training dynamics further support the design*).
> - **Clarity and Presentation**: (**R1**: *The manuscript is clearly written and easy to follow*), (**R3**: *Evaluation methodology is thoroughly reported, figures and Tables are well made*).
>
> We hope that our responses have fully addressed your concerns, and we welcome any further suggestions and discussions. **If our responses have resolved your concerns, we would sincerely appreciate your consideration for raising the score.**
>
>
> `References`
> ---
> ---
> [1] "A sober look at progress in language model reasoning: Pitfalls and paths to reproducibility." COLM (2025).
>
> [2] "Learning to reason under off-policy guidance." NeurIPS (2025).
>
> [3] "Learning What Reinforcement Learning Can't: Interleaved Online Fine-Tuning for Hardest Questions." arXiv:2506.07527 (2025).
>
> [4] "Thought-augmented policy optimization: Bridging external guidance and internal capabilities." arXiv:2505.15692 (2025).
>
> [5] "SFT memorizes, RL generalizes: A comparative study of foundation model post-training." arXiv:2501.17161 (2025).
>
> [6] "RL fine-tuning heals OOD forgetting in SFT." arXiv:2509.12235 (2025).
>
> [7] "On-policy distillation of language models: Learning from self-generated mistakes." ICLR (2024).

---

> > ### Comment · Reviewer_svi3 · 2025-11-23
> > **Response to author rebuttal**
> >
> > I commend the authors for their efforts in running additional experiments and analyses to answer my questions. Many of my original concerns have been resolved. However, there are still a few outstanding concerns:
> >
> > 1. Thank you for running the additional LLaMA-3.1B-8B experiment. This indeed showcases that the benefits of SRFT transfer also to non-Qwen models. However, I noticed that this experiment has been conducted on the base model and not the instruct model of LLaMA-3.1B, was there a specific reason for this? Indeed, performing GRPO on a base model might be ill-posed compared to an instruct variant. While I recognize it might be difficult to get this additional experiment in time before the discussion period ends, I would still request the authors to try to run a similar experiment with LLaMA-3.1B-Instruct and showcase the results.
> > 2. I agree with the points about GKD being difficult to implement given vocabulary mismatch. However, there do exist methods to conduct on-policy distillation under tokenizer mismatch, see https://huggingface.co/spaces/HuggingFaceH4/on-policy-distillation. As before, I recognize it might be difficult to run these experiments before the end of the discussion period, but I would definitely recommending having this experiment in the final version of the paper, since I believe it would strengthen the main claim of SRFT being a strong method for reasoning.
> >
> > Despite these points, I am raising my score to a 6 since many of my concerns have been addressed. However, I belive it would be important for the final version of the paper if the authors indeed are able to provide clarifications on these two points.

---

> ### Author Response · Authors · 2025-11-26
> **Thank you for the response**
>
> We thank you for engaging with us and for increasing your score! We sincerely appreciate your time and the valuable insights that have helped improve the quality of our work.
>
> - **Regarding Llama-3.1-Base vs. Instruct**: We appreciate you raising this point. We initially selected the Llama-3.1-Base model to align our experimental setting with recent related works in LLM reasoning and post-training[1][2][3], which primarily **utilize base models** for their experimentation. However, we fully agree that evaluating the Instruct variant is crucial for a complete assessment. We would like to highlight that we have verified SRFT’s effectiveness on both the **Base and Instruct versions** of the Qwen series (as detailed in **Table A1** of our appendix), where SRFT consistently outperformed baselines. This gives us confidence that the benefits will transfer to Llama-3.1-Instruct as well. As you kindly noted, it is challenging to complete these extensive training runs before the discussion period ends due to time constraints. Therefore, we commit to **conducting the Llama-3.1-Instruct experiments** and including these results in the final version of the paper.
> - **Regarding On-Policy Distillation with Tokenizer Mismatch**: Thank you for providing the resource regarding on-policy distillation under tokenizer mismatch. We agree that including this comparison will further strengthen the evaluation of SRFT. We will dedicate the necessary time to **implement OPD** and incorporate the comparative results into the final version of the manuscript.
>
> The constructive suggestions you gave during the rebuttal session are greatly helpful in improving the quality of our paper. Thanks for your time and hard work!
>
> `References`
> ---
> ---
>
> [1] "Learning to reason under off-policy guidance." NeurIPS (2025).
>
> [2] "Learning What Reinforcement Learning Can't: Interleaved Online Fine-Tuning for Hardest Questions." arXiv:2506.07527 (2025).
>
> [3] "Towards a unified view of large language model post-training." arXiv:2509.04419 (2025).

---

### Official Review · Reviewer_bDfv · 2025-11-01

**Soundness:** 2
**Presentation:** 3
**Contribution:** 2
**Rating:** 4
**Confidence:** 4

**Summary:**

This paper proposes SRFT (Supervised Reinforcement Fine-Tuning), a single-stage approach that unifies SFT and RL for LLM reasoning. The key idea is to balance demonstration distillation (SFT) and online exploration (RL) in one loop, using policy entropy as a dynamic indicator to weight the two objectives and avoid either overfitting to demos or premature entropy collapse. Conceptually, SFT makes coarse, global shifts in the policy, while RL makes fine, selective adjustments; combining them directly improves efficiency and stability. Empirically (Qwen-2.5-Math-7B), SRFT achieves improvement across five math-reasoning benchmarks over baseline methods like RL-only and SFT+RL. Training curves show faster reward growth, longer solutions, and stable entropy versus RL-only.

**Strengths:**

- The proposal to unify SFT and RL into a single stage is interesting and enables finer control over the trade-off between the two.
- The paper offers a comprehensive analysis of learning dynamics and the respective effects of SFT and RL for language models, improving our understanding of both paradigms.
- The manuscript is clearly written and easy to follow.

**Weaknesses:**

- The experiments are not yet fully convincing; more evidence is needed to demonstrate the effectiveness of the proposed SRFT method. In particular, the SFT data use DeepSeek-R1 responses (Line 360), which likely exceed the quality of the Qwen2.5-7B policy’s rollouts. Figure A2 suggests the model learns primarily from SFT—the SFT loss dominates—implying that distillation, rather than RL, drives most gains. For a fair comparison, both RL and SRFT should start from the same fine-tuned initialization, or the SFT data should be generated offline by the policy model itself to remove teacher-distillation effects.

- Prior work (e.g., OpenAI-o1, DeepSeek-R1) has demonstrated the importance of RL scaling for boosting LLM capabilities. Always incorporating SFT during training may hinder RL scalability and waste compute; the paper should consider when SRFT scales favorably versus when pure RL is preferable or how SRFT affects the scalability of RL.

- In the method part, the paper proposes various strategies, like entropy-guided reweighting and mixing different losses for SRFT. It would be better to provide detailed ablations to demonstrate the effectiveness of these designs.

**Questions:**

See weakness.

---

> ### Author Response · Authors · 2025-11-21
> **Authors Response (Part 1/2)**
>
> `Responses`
> ---
> ---
> >**Q1: Fair comparison between pure RL and SRFT.**
>
> **A1:**
> - **Meaning of Figure A2**: We apologize for the misunderstanding caused. Figure A2 is not intended to represent *the ratio of loss between demonstration data distillation* and *online rollout data RL*. Specifically, $w_\text{SFT}$ is used to balance the SFT and off-policy RL losses for demonstration data, while $w_\text{RL}$ is used to balance the positive and negative sample losses for rollout data. Therefore, the relative relationship between these two **does not represent** the relative magnitude of learning from demonstration data versus rollout data. Furthermore, according to Eq. (12) (Line 358), SRFT assigns **equal weight** to the learning losses of demonstration data and rollout data. We modify the relevant descriptions in the revised manuscript to clarify the meaning of Figure A2 to alleviate confusion.
>
> - **Fairness of Comparison**: The core advantage of the SRFT method is utilizing demonstration data and online rollouts to optimize the LLM policy **simultaneously**, whereas pure RL **cannot utilize** existing demonstration data. To demonstrate the effectiveness of the method, we compared SRFT in the original paper with three baseline methods—SFT, SFT$\rightarrow$RL, and SFT+RL—which used the **exact same DeepSeek-R1 demonstration data** across 8 benchmarks (Table 2). The results demonstrate that SRFT consistently achieves superior performance, attributable to its single-stage mechanism that effectively balances SFT and RL objectives. Therefore, we respectfully submit that these comparisons are fair and highlight the algorithmic advantages of our approach.
>
> ---
> >**Q2: How SRFT affects the scalability of RL.**
>
> **A2:** Thank you for your valuable insights. How SFT and RL affect LLM policy optimization is an important and underexplored problem [1][2][3][4]. In this work, we first analyze the roles of SFT and RL in LLM reasoning learning in Section 3, and then propose the SRFT method, utilizing adaptive weights based on policy entropy to balance the optimization process of SFT and RL. Within the experimental observation range of 0-500 steps, due to the design of adaptive weights, the policy entropy of SRFT consistently **remained at a higher level**, and performance **continued to improve** (Figure 6). Additionally, we ablate the adaptive weights to fixed values (Table 3), demonstrating the effectiveness of adaptive weights in the method. To investigate the impact on RL after removing SFT, we designed **a demonstration data annealing weight** based on Eq. (12):
>
> $\mathcal{L}\_{\text{SRFT}}(\theta) = \alpha(\mathcal{L}\_{\text{SFT}}^{\text{demo.}}(\theta) + \mathcal{L}\_{\text{RL}}^{\text{demo.}}(\theta)) + \mathcal{L}\_{\text{RL}}^{\text{self-rollout}}(\theta),$
>
> where $\alpha$ linearly decays from 1 to 0 over the first 150 or 300 steps. The results are shown in the table below:
>
> | Annealing Steps | AIME24 | AMC | MATH-500 | Minerva | Olympiad | Avg. |
> | :--- | :--- | :--- | :--- | :--- | :--- | :--- |
> | 150 | 31.4 | 69.5 | 86.2 | 36.1 | 53.8 | 55.4 |
> | 300 | 33.8 | 74.6 | 88.4 | 38.2 | 56.4 | 58.3 |
> | w/o annealing | 35.3 | 74.3 | 89.8 | 39.7 | 58.3 | 59.5 |
>
> The experimental results indicate that removing demonstration data after 150 or 300 steps affects performance improvement. We analyze that this is due to the quality of the demonstration data; for high-quality data like DeepSeek-R1, the behavior policy can be considered to be near the optimal policy. Therefore, in this situation, **continuously including SFT during the RL process is beneficial**. Furthermore, we believe that **demonstration data might affect RL performance improvement** in the later stages of training with longer steps. As this work is an attempt to combine SFT and RL in a single stage, set within 500 training steps, we suggest that discussing the relationship between RL scalability [5][6][7] and SFT under longer training steps can be further expanded as future research. We add the above results to the revised **Appendix A.3**.

---

> ### Author Response · Authors · 2025-11-21
> **Authors Response (Part 2/2)**
>
> >**Q3: Detailed ablations of various strategies in SRFT.**
>
> **A3:** Thank you for your suggestion. The initial manuscript ablated the two key mechanisms in SRFT, $w_\text{SFT}$ and $w_\text{RL}$, to fixed weights (Table 3), proving the impact of the adaptive weight mechanism on performance. We further supplement ablation experiments **removing key modules**, and the results are shown below:
>
> | Method | AIME24 | AMC | MATH-500 | Minerva | Olympiad | Avg. |
> | :--- | :--- | :--- | :--- | :--- | :--- | :--- |
> | w/ $\bar{w}_\text{SFT}$ (Fixed Weight) | 30.1 | 65.8 | 87.0 | 36.8 | 55.8 | 55.1 |
> | w/ $\bar{w}_\text{RL}$ (Fixed Weight) | 32.6 | 67.2 | 87.5 | 37.4 | 56.6 | 56.2 |
> | w/o $\text{demo}_\text{SFT}$ (Remove Loss) | 29.5 | 65.2 | 85.6 | 35.9 | 53.1 | 53.9 |
> | w/o $\text{demo}_\text{RL}$ (Remove Loss) | 30.8 | 68.0 | 86.4 | 36.2 | 54.5 | 55.2 |
> | SRFT | 35.3 | 74.3 | 89.8 | 39.7 | 58.3 | 59.5 |
>
> Here, w/ $\bar{w}\_\text{SFT}$ and w/ $\bar{w}\_\text{RL}$ ablate the adaptive weights to fixed weights of 0.1 and 1.0, respectively; w/o $\text{demo}\_\text{SFT}$ removes the SFT on demonstration data; and w/o $\text{demo}\_\text{RL}$ removes the RL on demonstration data. It is worth noting that since our method is built upon the RL framework (Line 279), the experiment excludes the ablation of $\mathcal{L}\_\text{RL}^\text{self-rollout}$, meaning online RL training is always included. The experimental results indicate that multiple components jointly contribute to the performance of SRFT, demonstrating the effectiveness of the method design. The above results and analysis are added to the revised **Section 5.2**.
>
> `Summary`
> ---
> ---
> Thank you for your valuable review and constructive feedback. We appreciate your recognition of our work's **interesting proposal**, **the fine-grained control over the trade-off between SFT and RL**, **our comprehensive analysis**, and **the clarity of the presentation**.
>
>
> We are also encouraged by the shared affirmations from Reviewers R2 (svi3), R3 (5rC4), and R4 (KnF6) regarding:
>
> - **Motivation and Novelty**: (**R2**: *The motivation to introduce the SRFT method is well-grounded*), (**R3**: *Motivation is clear*), (**R4**: *Methodologically clean and well-motivated*).
> - **Methodological Design**: (**R4**: *Grounded in an entropy-based view of SFT–RL integration*).
> - **Empirical Evidence and Analysis**: (**R2**: *Based on the empirical analyses*), (**R3**: *Evaluation on both in-distribution and out-of-distribution benchmarks is good... multiple ablations are presented*), (**R4**: *The evaluation is comprehensive, training dynamics further support the design*).
> - **Clarity and Presentation**: (**R2**: *The paper is well written and easy to follow*), (**R3**: *Evaluation methodology is thoroughly reported, figures and Tables are well made*).
>
> We hope that our responses have fully addressed your concerns, and we welcome any further suggestions and discussions. **If our responses have resolved your concerns, we would sincerely appreciate your consideration for raising the score.**
>
>
> `References`
> ---
> ---
> [1] "A survey of reinforcement learning for large reasoning models." arXiv:2509.08827 (2025).
>
> [2] "Model-free reinforcement learning from expert demonstrations: A survey." Artificial Intelligence Review (2022).
>
> [3] "RL fine-tuning heals ood forgetting in SFT." arXiv:2509.12235 (2025).
>
> [4] "On a connection between imitation learning and RLHF." ICLR (2025).
>
> [5] "ProRL: Prolonged reinforcement learning expands reasoning boundaries in large language models." arXiv:2505.24864 (2025).
>
> [6] "BroRL: Scaling reinforcement learning via broadened exploration." arXiv:2510.01180 (2025).
>
> [7] "The art of scaling reinforcement learning compute for LLMs." arXiv:2510.13786 (2025).

---

> ### Author Response · Authors · 2025-11-27
>
> Dear Reviewer,
>
> We are writing to follow up on our response to your valuable feedback. We hope that our clarifications, along with the revisions in the updated manuscript, have addressed your concerns. Assuming this is the case, we kindly ask that you consider re-evaluating your score. Otherwise, please let us know if you have any further questions.
>
> Thank you for your time and effort in reviewing our paper.
>
> Best regards,
>
> All Authors of Submission #1626

---

### Author Response · Authors · 2025-11-26
**Authors' Global Response (Before Discussion Freeze)**

This message has been entirely superseded by the **Authors' Final Summary**.

---

### Meta-Review · Area_Chair_2WYn · 2026-01-07

**Summary:**

This paper proposes SRFT, a single-stage framework that unifies supervised fine-tuning (SFT) and reinforcement learning (RL) for LLM reasoning via entropy-aware weighting. Reviewers generally found the motivation clear, the method conceptually interesting, and the empirical analysis thorough. Strengths highlighted across reviews include the unified training paradigm, detailed learning-dynamics analysis, comprehensive evaluation on in-distribution and OOD benchmarks, and generally strong presentation quality. Initial concerns centered on (i) fairness of comparisons and the role of high-quality demonstrations, (ii) lack of clarity in the token-distribution/entropy analysis, (iii) limited evidence beyond Qwen models, and (iv) insufficient ablations and discussion of stability, scalability, and computational cost. During rebuttal, the authors added substantial new experiments (including LLaMA-3.1 and AIME’25), clarified the visualization methodology, corrected reported numbers, expanded ablations, and discussed computational cost and training stability. One initially negative reviewer showed positive attitue to raise the score.

**Reviewer Concerns:**

Most of the concerns are addressed.

**Reviewer Scores:**

Reviewer bDfv: Likely 5-6.
Reviewer svi3: Explicitly raised from 4 → 6 after rebuttal.
Reviewer 5rC4: Remained at 6, leaning toward acceptance after new experiments.
Reviewer KnF6: 6, positive on methodology and empirical coverage.

---

### Decision · Program_Chairs · 2026-01-26

Accept (Poster)